# Global observations of reflectors in the mid-mantle with implications for mantle structure and dynamics

Lauren Waszek[1,2], Nicholas C. Schmerr[3] & Maxim D. Ballmer [4]

Seismic tomography indicates that flow is commonly deflected in the mid-mantle. However, without a candidate mineral phase change, causative mechanisms remain controversial. Deflection of flow has been linked to radial changes in viscosity and/or composition, but a lack of global observations precludes comprehensive tests by seismically detectable features. Here we perform a systematic global-scale interrogation of mid-mantle seismic reflectors with lateral size 500–2000 km and depths 800–1300 km. Reflectors are detected globally with variable depth, lateral extent and seismic polarity and identify three distinct seismic domains in the mid-mantle. Near-absence of reflectors in seismically fast regions may relate to dominantly subvertical heterogeneous slab material or small impedance contrasts. Seismically slow thermochemical piles beneath the Pacific generate numerous reflections. Large reflectors at multiple depths within neutral regions possibly signify a compositional or textural transition, potentially linked to long-term slab stagnation. This variety of reflector properties indicates widespread compositional heterogeneity at mid-mantle depths.

[1] Department of Physics, New Mexico State University, 1255 North Horseshoe, Las Cruces, NM 88003, USA. [2] Research School of Earth Sciences, The Australian National University, Canberra, ACT 0200, Australia. [3] Department of Geology, University of Maryland, 8000 Regents Drive, College Park, MA 20742, USA. [4] Institute of Geophysics, ETH Zurich, Sonneggstrasse 5 8092 Zurich, Switzerland. Correspondence and requests for materials should be addressed to L.W. (email: lauren.waszek@cantab.net)

The Earth's mantle undergoes significant mineralogical and physical changes as temperature and pressure increase with depth. Characterising these changes in the upper 400–800 km has advanced our understanding of heat and material fluxes through the mantle. In particular, variations in the depth of seismic discontinuities, which reflect and convert seismic waves[1, 2], have been used to map solid-to-solid mineralogical phase changes and thus regional variations in mantle temperature and/or composition. A classic example is the pressure–temperature sensitivity of the depth of major discontinuities that bound the Mantle Transition Zone (MTZ) at 410 and 660 km. These boundaries demarcate transitions of olivine to wadsleyite and ringwoodite to bridgmanite+ferroperclase[3]. In contrast, there are no known phase changes in mantle minerals that readily explain regional discontinuities (here termed seismic 'reflectors') at mid-mantle depths (from 800 to 1300 km)[4–12]. Thus the origin and geodynamic implications of these mid-mantle reflectors remain elusive.

Recent work posits that the mid-mantle may represent a significant transition in Earth's rheology and/or composition[13–15]. Tomographical studies have found that only few recently subducted slabs sink unimpeded through the MTZ[16, 17]; many slabs flatten and appear to stagnate at either ~660 km or ~1000 km depth[18]. Upwelling mantle plumes also commonly show deflection at similar mid-mantle depths[19, 20]. However, observations of Tethys and Farallon lithosphere in the lower mantle[21] reveal that flow crosses these depths, at least regionally.

While deflections of mantle flow near 660 km depth can be related to the effects of a major phase transition[22, 23], those in the mid-mantle have instead been ascribed to a range of alternative mechanisms. These include the presence of a viscosity jump[14, 24], radial change(s) in mantle composition[13] and mineral phase changes for particular material compositions, such as transitions within subducted slabs[25, 26] and/or impedance contrasts arising from the different composition of the subducted material itself[27, 28]. Testing various processes for the origin of the reflectors (compositional vs. rheological) requires detailed evaluation of seismic reflections on a global scale.

Previous work shows that any mid-mantle reflectors display immense variation in seismic properties and depths, and no global mantle discontinuity has been detected beneath the 660. Abundant regional mid-mantle reflectors and scatterers occur from 700 to nearly 2000 km depth in the mantle[4–11, 13, 29]. Reflectors are observed beneath areas of active subduction including Indonesia and the Marianas[6–12]. Numerous small-scale (~10s of kilometres) features are detected around the Pacific Ocean, which are interpreted as subducted oceanic material[27–33]. Studies also find evidence for reflectors in regions of upwelling, such as the Hawaiian and Icelandic hotspots[19, 34, 35]. Further isolated observations are situated well away from subduction zones and hotspots[12, 36–38]. There are also several locations where mid-mantle reflectors have not been found despite detailed examination, such as the Tonga subduction zone[2, 39], and vast regions remain to be mapped at mid-mantle depths[13]. This is in part due to a lack of studies of the mid-mantle on a global scale. Indeed, a comprehensive worldwide investigation is required to further our understanding of the mid-mantle.

Here we perform a systematic global-scale seismic interrogation of mid-mantle reflectors. We search for reflectors in the 800 to 1300 km depth range, using precursors to the seismic phase SS. This shear wave has two paths in the mantle and reflects once from Earth's surface at its midpoint; SS precursors are generated by any reflectors beneath the surface. The arrival time of this seismic phase is thus sensitive to the depth of the bounce point. A large global dataset of SS-precursor arrivals is partitioned into regional bins and stacked into vespagrams, employing common

mid-point stacking (see Methods section for more information). We demonstrate using synthetic modelling that our dataset is sensitive to near-horizontal reflectors with length scales on the order of 500 to 1500 km and show that the reflectors are too small to be resolved by global tomography techniques. We measure the location, geographic size, depth, and impedance contrast of the reflectors in the mid-mantle, finding large variability. We investigate a range of different geographical bin sizes to constrain the variation in these properties across multiple length scales and perform more detailed analysis in regions of higher data sampling density. Reflector properties are evaluated in the context of average seismic velocity from global tomography models[40]. Such an evaluation puts our observations into the framework of global mantle flow patterns[41], performed to improve our understanding of variations in mantle temperature and composition (e.g., refs. [16, 17, 20]). Mapping reflectors in the mid-mantle is key to constraining the heterogeneity that may exist in the mid-mantle, with implications for the history of mantle mixing.

## Results

**General observations.** A systematic search reveals widespread regional reflectors in the mid-mantle (Figs. 1, 2a–c and 3a, Supplementary Figs. 1–8). The wide geographic variation in the depth and lateral extent of these reflectors indicates that a coherent global discontinuity at any individual mid-mantle depth can be excluded; correspondingly, a global stack shows no features here (Supplementary Fig. 1). This is consistent with global seismic velocity models[42, 43]. Reflectors occur across the entire depth range explored, corroborating previous regional studies[2, 4–8, 11]. Reflections from 875 km depth are most abundant (Fig. 4a), with less pronounced peaks in the range of 1000–1300 km depth. The geographically most extensive reflectors are located beneath the Pacific Ocean and (offshore) eastern South America. The scale lengths of reflectors vary laterally over 500–2000 km, and some bin locations have multiple reflectors at two or more depths (Fig. 2c). Reflectors of small regional extent (<500 km) are located beneath the North Pacific, western South America, and Eastern Europe, in agreement with prior studies (e.g., refs. [9–12, 32]).

Most precursors have the same polarity as the amplitude of SS, implying either a velocity or density increase with depth (i.e., a positive shear impedance increase with depth), but a subset (26%) of the observed reflectors display opposite polarity (Figs. 3b and 4b). Polarities of the reflectors are consistent within geographic regions but do not vary systematically with depth (Supplementary Fig. 8). The corresponding shear-wave velocity contrasts (assuming no density contrast) range from $\pm 0.7$ to $\pm 3.2\%$ (Fig. 4b); contrasts below 0.7% are too weak to be detected by our stacking methodology. Likewise, density contrasts (assuming zero contrasts in intrinsic shear modulus) are therefore calculated as approximately $\pm 1.4$ to $\pm 7.3\%$. Actual shear impedance contrasts will be intermediate combinations across the two properties and also depend upon the geometry of the reflector within the bin; the observed seismic properties represent an average across the length scale of the bin.

Several areas are characterised by the absence of reflections in our stacks across the full depth range explored, termed here as 'non-detections' (Fig. 2d). Indeed, 24% of the bins in Fig. 3a do not display reflectors (Table 1). Notable coherent geographical regions without reflectors exist beneath the North Pacific (Aleutian Trench), central Europe, and the Brazilian coast (Peru-Chile Trench). The amount of non-detections varies regionally. Some areas display a higher proportion of bins with coherent reflectors (e.g., beneath the Pacific Ocean), whereas other locations have a higher percentage of bins with non-

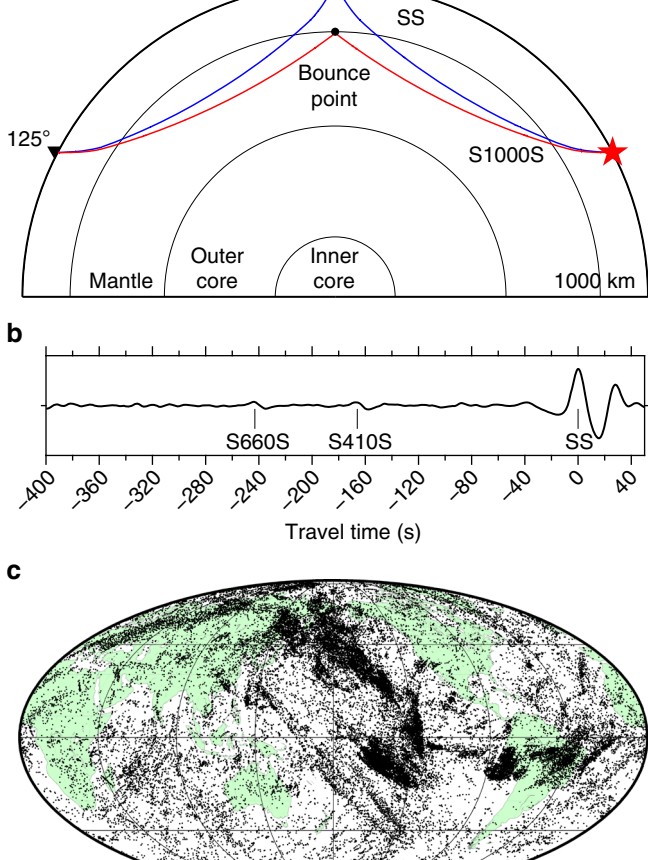

**Fig. 1** SS data and coverage. **a** SS and S1000S ray paths. **b** Example of a high-quality SS seismogram (event: 1 January 2016, Indian-Antarctic ridge; station: DLRN). Precursors from 410 and 660 km are clearly visible. **c** Global dataset showing 45,634 SS bounce points

detections (e.g., beneath Europe and the Brazilian coast) (Figs. 5 and 6). The presence and quantity of non-detections also varies across length scales, evidenced by variation between bin sizes (Supplementary Figs. 9, 10). Larger bins typically display a higher proportion of non-detections (Supplementary Table 1).

A lack of reflector may result from multiple factors, not solely limited to the absence of sub-horizontal mid-mantle heterogeneity. For example, small impedance contrasts that do not generate energy above the noise level, gradual radial transitions (>65 km) including gradual thermal gradients that do not produce reflectors at SS frequencies, or complex three-dimensional (3-D) structure that does not stack coherently within the bin would not generate reflectors[44–46]. Owing to the mid-point stacking technique, any reflectors that are not oriented sub-horizontally, such as dipping structures, will also not stack coherently.

We examine these averaging effects across bin sizes, by comparing the small and large bins to confirm variation in reflector coherency across lateral length scales (Supplementary Figs. 4, 5). The averaging effect is exemplified beneath the mid-Pacific Ocean, where the depths of reflectors vary significantly for the 5° bins (Supplementary Fig. 4a). Conversely, the larger 15° bins predominantly display fewer reflectors (Supplementary Fig. 4d); a consequence of averaging over small length-scale variations. We also find bins with non-observations that are situated directly adjacent to bins with robust detections, despite

using overlapping bin geometries. For example, bins with no reflectors are located within regions of significant variability beneath the South Pacific. This observation suggests highly complex structure that is not fully resolved by SS precursors and could be constrained by alternative, higher-resolution techniques (e.g., refs. [27], [33], [35]).

**Observability of reflectors**. As mentioned above, any S-wave reflections retrieved by our method require sub-horizontal reflectors of a particular impedance contrast and lateral extent for a given bin size. We observe precursor/SS amplitude ratios in the range of ±0.03, and the smallest SdS/SS amplitude ratios that we detect are 0.0065. This marks the approximate limit of detectability of the precursors; precursor signals that are smaller than this amplitude will not be visible above the noise level. Below, using synthetic modelling, we quantify the sensitivity of the SS precursors to the sizes and strengths of reflectors for multiple bin sizes. This allows us to establish a framework for the interpretation of our observations across different length scales and place constraints on the limitations of the method.

We use the 2.5-D spectral elements code AxiSEM[47] to generate synthetic seismograms and stacks and obtain estimates on the observability of reflectors as a function of their strength and size relative to the bin. We determine candidate seismic impendence contrasts to which our observations correspond and explore the influence of the lateral size of reflector as a function of bin size. We present these as contour plots (Fig. 7), which reveal the detection limits as a function of bin size (yellow-to-red colours in Fig. 7). We test the same size and strength of reflectors for bins of radii 25°, 15° and 10°, in order to also explore the dependence of observability of a given reflector on absolute bin size.

The modelling reveals that, generally, the SS precursors are sensitive to horizontal structures consistent on length scales similar to the bin sizes (500–1500 km), with detectable reflectors resulting from sharp and large transitions in shear impedance (<5 km gradients, shear impedance <5%). As expected, reflectors that comprise a larger proportion of the bin area are detectable for much lower velocity contrasts than smaller reflectors. Larger reflectors will generate coherent signals in stacks, whereas smaller reflectors will be somewhat suppressed by bounce points from portions of the cap with no signal, reducing observability of the SS precursors.

The influence of reflector size with respect to the bin is clearest in Fig. 7a, where the relative size of the reflector proportional to the bin increases from 10 to 50%. Putting this into the context of our observations, the smallest observed SdS/SS amplitude ratio of 0.0065 corresponds to a minimum impedance contrast of about 0.8%. Thus the synthetic calculations show that a reflector at this limit of observability will be observed in a bin for which it comprises at least 50% of its size. In other words, the weakest reflector we detect has to be on the order of 500 km in size.

As bin size decreases, the observability of reflectors is skewed significantly towards detecting smaller reflectors. For example, for bin sizes of radius 10°, almost all theoretical reflectors may be observed in high-quality stacks. This is corroborated in our observations for various bin sizes, whereby the proportion of observations generally increases with decreasing bin size (Table 1). Thus the reflectors that are only resolvable in the smaller bin sizes must vary on short length scales and hence are suppressed within larger bins. This confirms that the method is primarily suited to detecting features on the length scales of the bins. The modelling thus allows us to estimate the geographical size as well as lateral variation in topography of the reflectors in our subsequent analysis, based on any consistent variation across bin sizes (or lack thereof).

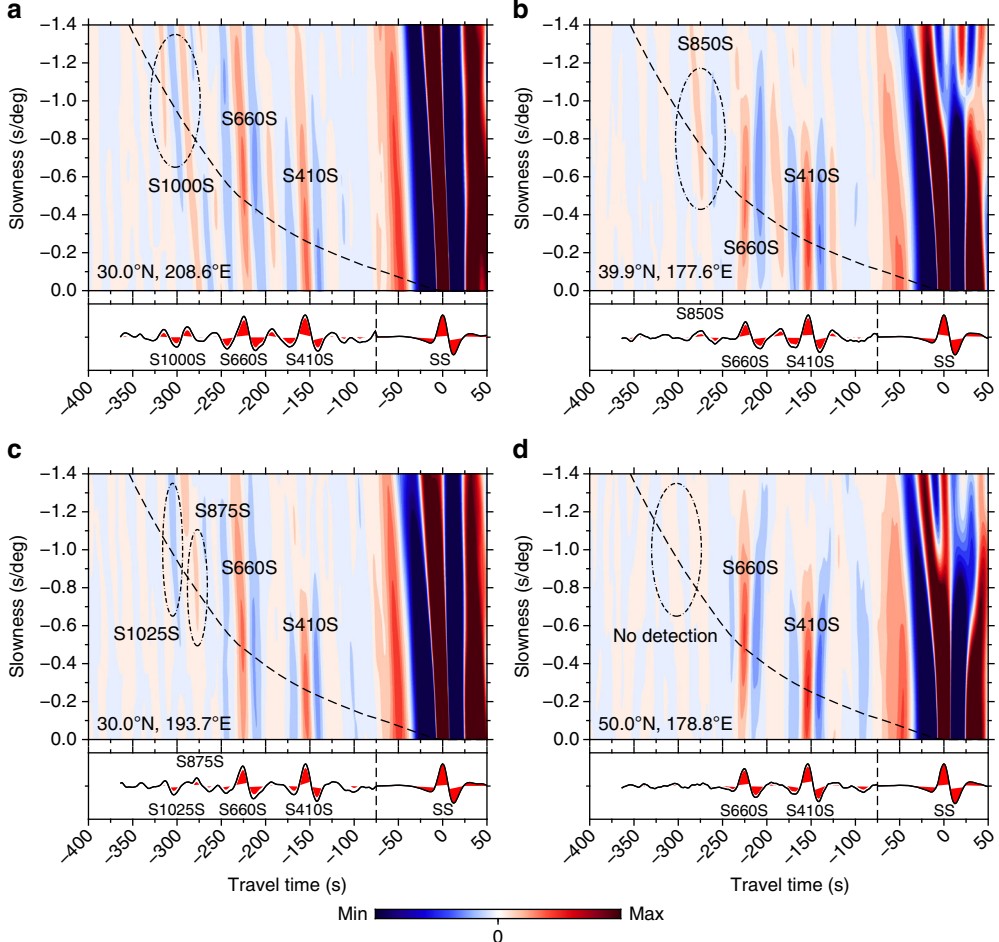

**Fig. 2** Examples of vespagrams from 10° bins. **a** Negative polarity reflector (number of records, NR = 1492). **b** Positive polarity reflector (NR = 1486). **c** Multiple mid-mantle reflectors (NR = 1348). **d** No mid-mantle reflectors (NR = 1647), a non-detection. Areas of colour represent arrivals of coherent seismic energy from bootstrap resampling; colour scales normalised to SS amplitude. Slownesses are relative to the SS slowness of 12.56 s/° at 125°. Cross-sections are taken through the predicted arrival time and slowness of SS precursors (dashed line). Red peaks are 95% confidence levels

Our calculations highlight the trade-off between reflector size and impedance contrasts for a bin. The measurements represent an average value across the size of the bin, and it is not possible to distinguish between large but weak reflectors versus small but strong reflectors within an individual bin. Consequently, in terms of amplitude ratios, we consider only the polarity rather than absolute measurements. However, the lateral variability of amplitude ratios may be used to infer lateral variation in strength as well as the presence of a reflector (e.g., in the case of a laterally intermittent discontinuity).

In future, more computationally intensive modelling work, as well as more data with different length scales of resolution, is required to investigate the complex features that exist in the mid-mantle. Our synthetic tests elucidate that we should expect averaging across any structures present in the bin. Very likely, such structures include multiple reflectors at different depths in one bin, reflectors with laterally varying or potentially anisotropic impedance contrast, as well as tilted reflectors.

**Relationship to 3-D tomography**. We explore the relationship of reflectors to radial seismic velocity gradients, and the influence of 3-D velocity structure in the mantle, to explore various potential structural and geodynamical origins for the reflectors. Investigations are performed for two recent shear-wave mantle tomography models, S20RTS[48] or SEMUCB-W1[49]. For each model, we

calculated the average 3-D radial velocity gradient for a bin within ±25 km of the estimated depth of the reflector. The SS precursor data are sensitive to velocity gradients that occur across this radial length scale or less. We identified no robust correlation between reflectors and velocity gradients (Fig. 8), indicating that the mantle structures that cause the reflections are not resolved by tomography. Notably, all calculated shear-wave velocity gradients are positive, yet a significant proportion of the reflectors have negative impedance contrasts. For both of these reasons, the reflectors must therefore result from structures with shorter length scales than those in the tomographic models.

Lateral velocity anomalies as resolved by mantle tomography may also affect the localisation of reflectors by SS precursors. Our initial dataset was not corrected for 3-D velocity structure, and we test the influence of 3-D heterogeneity by performing corrections for individual ray paths, by calculating for delay times of S1000S with respect to SS. We find that there are limited travel time differences between the vespagrams with uncorrected data and vespagrams corrected for each model (Supplementary Figs. 11, 12). For all of the 10° bins, S20RTS alters the times by an average of 0.6 ± 3.3 s, whereas the average change for SEMUCB-W1 is 2.3 ± 2.8 s (using the standard deviation of all corrected measurements as the error). This corresponds to depth corrections and errors of approximately 3 ± 17 km and 12 ± 14 km. This is likely due to averaging effects over the range of distances and azimuths within each bin (see Supplementary Fig. 13 for

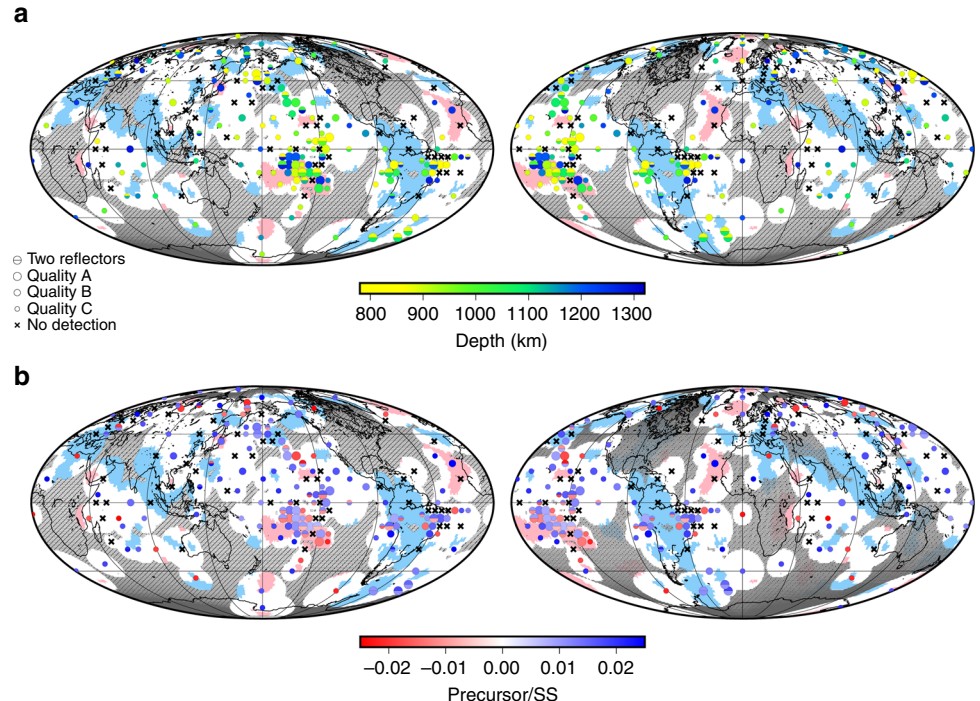

**Fig. 3** Depths and impedance contrasts of observed mid-mantle reflectors. **a** Discontinuity depths, with observations (circles) and bins with non-detections (crosses). Symbols are plotted at the average geographic location of bounce points within each bin, and circle sizes are scaled by quality of observations. Bins without sufficient coverage are omitted (Supplementary Fig. 7). Superimposed are average tomographically fast (blue) and slow (pink) regions at 1100 km depth[40]. Hatched regions are those with no data coverage from the bins shown. **b** Corresponding plot showing precursor/SS amplitude ratios. The maps are populated iteratively, beginning with the smallest bin sizes, to produce the highest possible data coverage (see Methods section). The data coverage (non-hatched areas; see also Fig. 5, Supplementary Figs. 4 and 5) provides an indicator as to the sensitivity of each bin

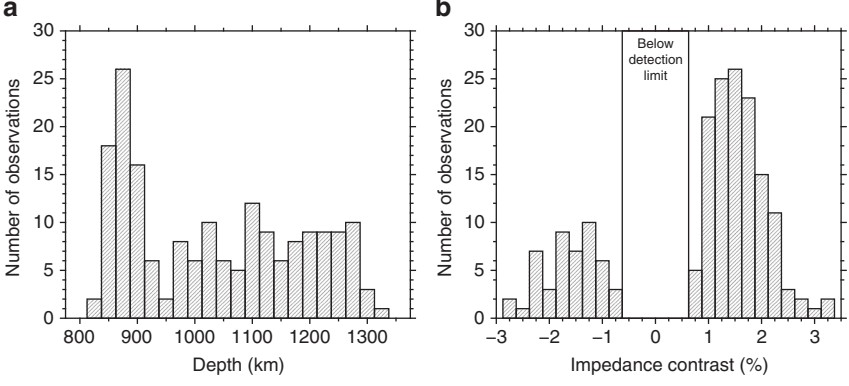

**Fig. 4** Depths and impedance contrast of observed mid-mantle reflectors. **a** Histogram of discontinuity depths showing bins of all sizes. **b** Histogram of impedance contrasts estimated from shear-wave velocity contrasts. For corresponding histograms for all bin sizes, see Supplementary Figs. 6 and 7

distribution across all data). The major effect of the corrections is in the waveform shape of the precursors, rather than significant differences in their arrival time. These discrepancies may result from defocussing of reflectors at other depths, as well as the main SS arrival, and influences the travel time of the maximum amplitudes. The difference is clear in the shape of the waveforms in the cross-sections and particularly noticeable for the SEMUCB-WM1 corrections (Supplementary Fig. 12c). As a consequence, we do not use 3-D corrections for our data analysis, as the average correction falls below the extent of our 25 km depth bins.

**Regional domain analysis of reflectors.** We interpret our observations in the context of mid-mantle tomography models

(Fig. 3). Data from a recent study integrate cluster analysis of five mantle tomography models to independently generate 'vote maps' of seismically fast, slow, and neutral (i.e., with velocities close to the global average) domains[40]. Each tomography model is allocated a 'vote' as to whether the mid-mantle structure is grouped into one of these three clusters (or domains) to generate a global combined map. We define each bin according to the average votes, which accounts for bins that may incorporate multiple domain types. Fig. 6 shows cross-sections through these vote maps; shades of blue and red indicate regions for which the majority of tomography models agree that mantle rocks are fast and slow, respectively; no shading corresponds to 'neutral'.

**Table 1 Quantity and percentage of reflector observations and polarity**

| Cap size | Domain | Fast | | Neutral | | Slow | |
|---|---|---|---|---|---|---|---|
| Combined caps | Observations | 209[n,s] | 71% | 571[f,s] | 77% | 130[f,n] | 85% |
| | Non-detections | 84[n,s] | 29% | 173[f,s] | 23% | 23[f,n] | 15% |
| | Positive polarity observations | 158[s] | 76% | 422 | 74% | 90[f] | 69% |
| 5° caps | Observations | 113[s] | 73% | 309[s] | 75% | 83[f,n] | 88% |
| | Non-detections | 42[s] | 27% | 102[s] | 25% | 11[f,n] | 12% |
| | Positive polarity observations | 82[s] | 72% | 209 | 68% | 50[f] | 60% |
| 7.5° caps | Observations | 97[s] | 77% | 263[s] | 75% | 50[f,n] | 68% |
| | Non-detections | 29[s] | 23% | 87[s] | 25% | 24[f,n] | 32% |
| | Positive polarity observations | 68[s] | 70% | 197 | 75% | 40[f] | 79% |
| 10° caps | Observations | 69[n] | 52% | 219[f] | 60% | 42 | 58% |
| | Non-detections | 62[n] | 48% | 147[f] | 40% | 31 | 42% |
| | Positive polarity observations | 48 | 70% | 164 | 75% | 32 | 77% |
| 15° caps | Observations | 79 | 57% | 155 | 53% | 26 | 53% |
| | Non-detections | 59 | 43% | 137 | 47% | 23 | 47% |
| | Positive polarity observations | 59 | 74% | 104 | 67% | 17 | 67% |

Proportion of cluster votes for caps with observations, proportion of votes for caps with no reflectors, and proportion of observations with positive polarity, for each type of seismic velocity domain. Corresponding fractions are also included as percentages; for each bin size, percentages of observations and non-detections (row 1 plus row 2) therefore total 100%. Results are shown for the combined bin sizes (i.e., the map completed iteratively beginning with smallest cap size) and for all the bin sizes separately. Variations between bins of different sizes are related to the lateral scale length of reflectors. The domains are determined by calculating the average cluster analysis votes across the bin at the depth of the reflector (see Methods section for details). Significant differences between domain types, calculated via z-tests, are shown via the inclusion of superscript letters, taking the first letter of each domain type. The presence of a superscript letter indicates that the domains differ at the $p = 0.1$ level. Full $p$-values are presented in Supplementary Table 1.

We analyse our data in the context of the domain in which reflectors are located, since these roughly correspond to the degree-2 structure of whole-mantle convection also predicted by global geodynamic models[41]. Fast regions are commonly related to downwelling cold material (subducted slabs), whereas slow regions correspond to hot upwellings (plumes). Neutral domains are not correlated to either upwelling or downwelling flow and may be characterised in some regions by the impedance of radial flow, such as stagnation of slabs or plumes at various depths in the MTZ[17, 20, 50]. By considering our observations in the context of the average seismic velocity properties, we obtain an insight into the relationship of horizontal reflectors to mantle flow and deflection processes and associated thermochemical heterogeneities.

We find statistically significant differences between seismic domains for bin sizes up to 10° ($p < 0.1$; i.e., the probability of different domains having the same seismic properties is <10%) (see Methods section for full details). We characterise each bin according to the average seismic domain votes and use a z-test to perform systematic statistical comparisons for proportion of reflector observations versus non-detections and polarity between each domain types (Table 1). For the combined bin approach and 5° bins, the proportions of bins containing reflectors differ significantly between seismic domains ($p < 0.05$). As bin size increases, both the differences between domains and the significance decrease and are ultimately no longer statistically significant at 15° bin sizes. This statistical analysis highlights the averaging effects for larger bins, including the fact that larger bins are more likely to encompass multiple seismic domains.

Geographical bins from slow domains (upwellings) predominantly show reflectors (85%), which vary on short length scales (500 km) across the full depth range, and roughly follow the tomographically defined domain boundaries in vertical cross-sections (Fig. 6a, c). The small length scales of lateral variations are highlighted by the decrease in the number of reflectors observed as bin size increases. This reveals that the reflectors vary on length scales corresponding to the size of the 5° bins (up to 1000 km) and thus are not resolvable in the larger bins. Of the reflectors detected, relatively more possess negative polarity (31%) compared to other domains, suggesting local mantle heterogeneity to produce such seismic structures[51]. This supports the inference that these reflectors correspond to a significant compositional and/or structural difference between slow regions and other seismic domains (see Table 1).

In contrast, fast domains (downwellings) are characterised by relatively more non-detections than slow regions (i.e., only 71% of the reliable bins contain reflectors). Spatially coherent reflectors are rarely found within the bulk of the fast domain, and there is no consistent relationship to length scale of observation. The majority of reflectors are located near to the edges of the domains (Figs. 3a and 6a, b). Comparisons between bins of differing sizes reveal no trend in quantity of detections with increasing bin size, indicating sporadic, isolated reflectors, with varied length scales across our range of resolution. In comparison to the slow domains, a greater proportion of the observed reflectors have positive polarities than negative (76%).

Compared to fast and slow regions, neutral domains contain an intermediate proportion of reflectors within bins (77%), with the majority exhibiting positive polarity (74%). An assessment of the proportion of observations for different bin sizes in the neutral domains reveals that lateral scale lengths of the reflectors are geographically consistent across larger length scales than other domains. The defining characteristics of reflectors in neutral domains, compared to those in fast and slow regions, is that they are often laterally coherent across bins, forming very large and continuous features with consistent depths. Neutral regions further display a majority of shallow detections around 900 km depth; 50% of the reflectors are within ±100 km of this depth. Unlike in fast and slow domains, these reflectors tend to be situated away from domain edges and can extend across the entire domain.

## Discussion

The observed mid-mantle reflectors do not exhibit any geographic relationship to surface features. Instead, they correlate to mid-mantle structure as independently imaged by seismic tomography. There is a good agreement between tomography models in terms of the locations and extent of mid-mantle tomographic domains[40], which reflect large-scale mantle flow patterns[41]. For example, broad mid-mantle upwelling is likely manifested above the large low shear-velocity provinces (LLSVPs) of Africa and the South-central Pacific. Downwelling should be focussed along the high velocity belts found across Asia and the

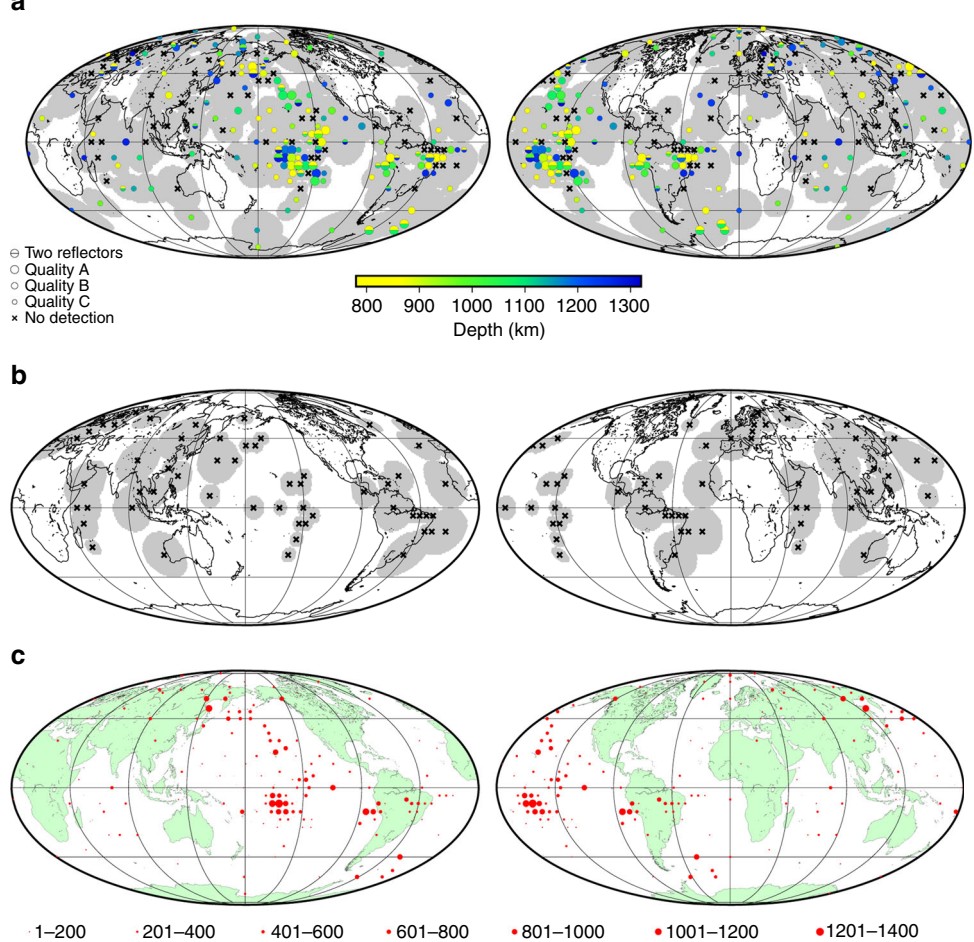

**Fig. 5** Data coverage and distribution of bounce points with respect to bins. **a** Data coverage of all bins. Shaded regions correspond to regions included in bins. Non-shaded areas are regions with insufficient data coverage or poor-quality stacks that were removed after quality checking. **b** Data coverage for only bins with non-detections. **c** Data distribution per bin. The size of circles in each bin shows the number of data in each bin (see legend). Corresponding maps for all bins sizes are in the Supplementary Material, along with maps showing the regions without data coverage superimposed on the combined maps

Americas, where Tethys, Pacific and Farallon lithosphere sinks through the mid-mantle[16].

The South Pacific is our best-resolved example of a slow region, with dense horizontal reflectors that vary on short lateral length scales. Reflectors are absent in only very few slow domain bins (primarily beneath the Pacific Ocean and likely as a result of variation on length scales too small to resolve) and occur near the edges or tops of slow domains (Fig. 6) or LLSVPs. One hypothesis for mid-mantle reflections is that they result from a compositional change across the top edges of the LLSVPs, which are interpreted as thermochemical piles that host primordial material and/or basalt-enriched subducted material[52–54]. Some thermal contribution may arise if the gradients are extremely strong (occurring over vertical distances of less than approximately 65 km). The abundance of reflectors with near-equal occurrences of positive and negative impedance contrasts may attest to heterogeneity within the LLSVPs[55]. The top of the low-velocity anomaly would produce a negative impedance contrast, although such a feature may be gradational. Streaks of basalt/harzburgite would produce alternating bands of elevated and lowered seismic velocity and density contrasts, similar to the observed varied impedance contrasts and polarities within the data. This interpretation implies that the numerous reflectors within the seismically slow region map the shallow roof of a compositionally distinct Pacific LLSVP[40, 51, 55] (Fig. 6). Although not recovered here due to sparse

data coverage in the region (Fig. 5a, Supplementary Fig. 9), we would predict similar reflectors near the roof of the African LLSVP.

The comparatively high quantity of non-detections in 'fast regions' is partially due to sparse data coverage in regions of expected mid-mantle downwellings (Fig. 5a, Supplementary Fig. 9), although the best-example fast regions in Europe have extensive data sampling (Fig. 3). Heterogeneity in fast regions (i.e., downgoing slabs of cold, seismically faster basalt and harzburgite) is expected to be dominantly sub-vertically oriented, as well as small scale, and thus difficult for the SS precursors to resolve compared to shorter wavelength methods[4–11]. Reflectors smaller than ~500 km are difficult to be resolved (see Fig. 7). Alternatively, no reflectors would be detected if the impedance contrasts are small (less than approximately 0.7% averaged across the entire bin). The scattered mid-mantle reflectors in these regions are consistent with small-scale heterogeneity on the order of a few hundred kilometres, as may be expected from the vast range in composition of subducted material.

Deeper sub-horizontal reflections as observed from within the fast domains may arise from coherently stacked piles of basalt[56] (Fig. 9). Alternatively, they may be generated by phase transitions within the basalt, continental crust, or sediment layers of the subducted slabs[26, 57]. Such an explanation requires specific geometries of the these (thin) layers to sustain large (>500 km)

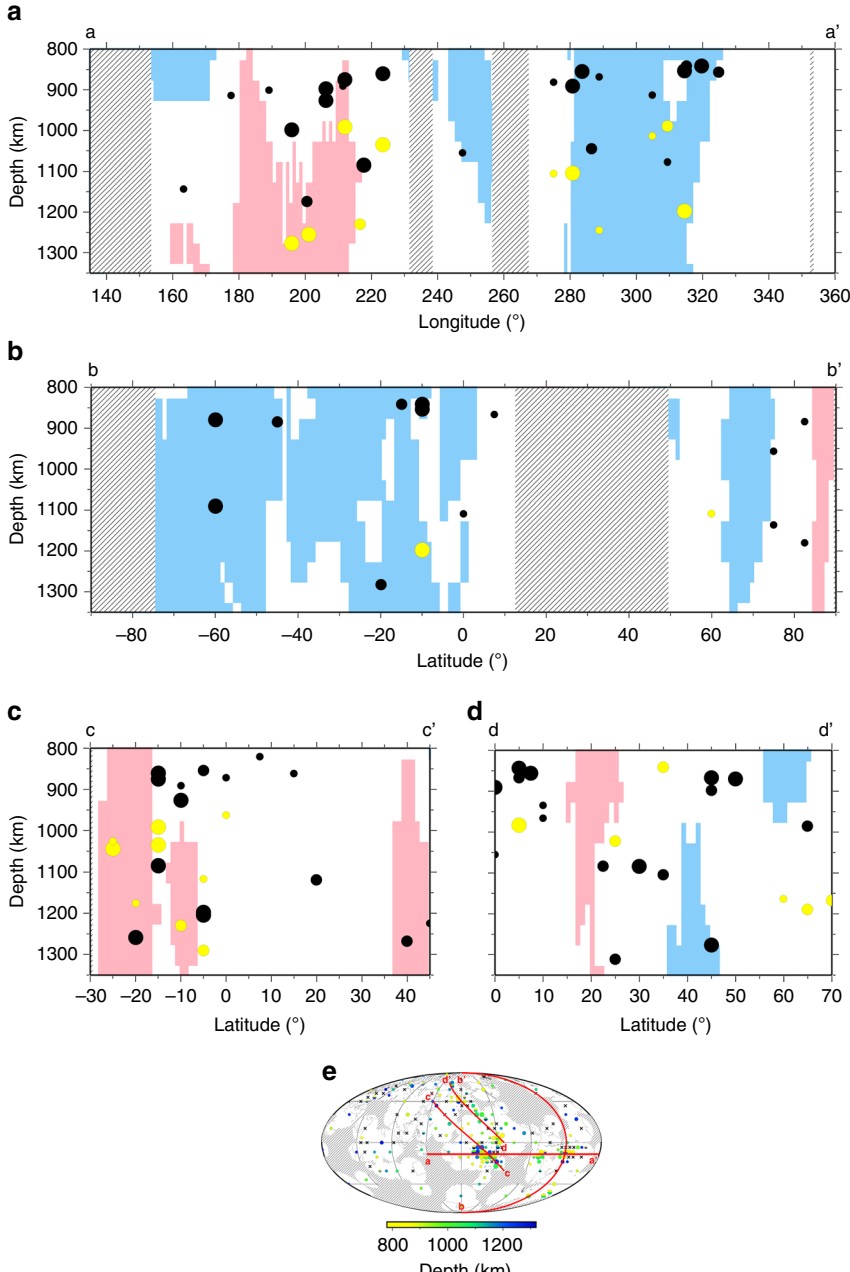

**Fig. 6** Cross-sections through vote maps for five mantle tomography models across four regions of high data density. **a** Central Pacific and South America (−12°N, 135°E to −12°N, 0°E). **b** North and South America (90°N, 315°E to −90°N, 315°E). **c** Central Pacific Ocean (45°N, 135°E to −30°N, 240°E). **d** North Pacific Ocean (70°N, 150°E to 0°N, 235°E). **e** Map showing locations of cross-sections. Observations within 1000 km lateral distance of each cross-section are included, superimposed at their calculated depths. Tomographically fast and slow regions are shown in blue and pink, respectively, calculated at every 50 km depth, where three or more seismic tomography models agree in cluster vote analysis[40]. Unshaded regions are neither fast nor slow. Hatched areas correspond to locations with SS precursor coverage. Black circles are positive polarity reflectors; white circles are negative polarity. Note the vertical exaggeration of the depth slices

coherent reflectors, which would generate a reflector with or without an accompanying phase transition. We predict similar results in other fast regions (e.g., Central America), to be obtained with methods of higher spatial resolution than SS (e.g., refs. [11, 33, 35]). The observations near to the edges of the fast regions are likely generated by the expected large impedance contrast between compositionally distinct domains.

In our best-example 'neutral region' in the Northeast Pacific, there are two dominant geographically large reflectors at 850 and 1050 km, with scattered deeper detections (Fig. 6). Possible mechanisms for the deeper reflections are regional changes in

rock texture or composition with depth[13], such as a transition from pyrolite to bridgmanite-enriched mantle[50, 58]. Our reflections could alternatively correspond to a regional jump in viscosity, which has been proposed to occur at mid-mantle depths[14]. Shallower reflections may arise from the top and/or bottom of a thermally equilibrated (i.e., fossil) slab that stagnates atop the (textural or compositional) viscosity jump[14, 50]. Long-term stagnation can occur as slab sinking is impeded at a viscosity (or density) contrast to allow progressive slab warming that removes the negative buoyancy of the slab. Once oriented horizontally, a slab then becomes detectable by the SS precursor data. The

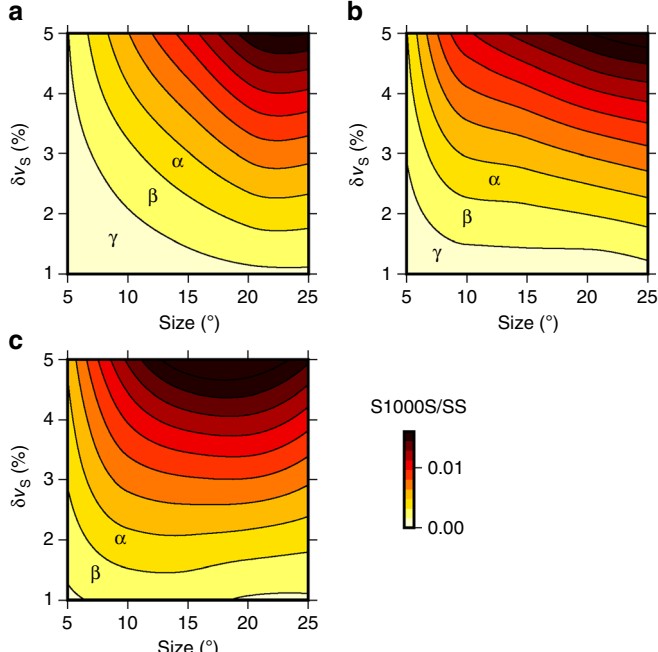

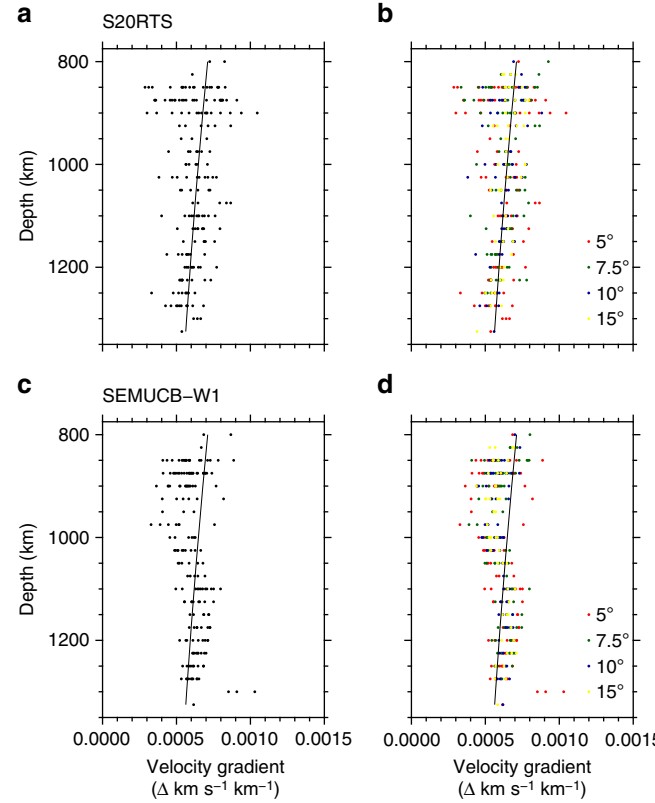

**Fig. 7** Theoretical S1000S/SS amplitude ratios for reflectors as a function of lateral size and shear-wave velocity contrast. **a** 25° bins. **b** 15° bins. **c** 10° bins. Reflectors are introduced at 1000 km depth and centred on the SS bounce point, for an event-receiver epicentral distance of 125°. The contours correspond to the S1000S/SS amplitude ratio, as measured in cross-sections through the stacked vespagrams. Contour γ marks the realistic limit of observability. Contour α is not detectable, and Contour β is observable only in extremely high-quality data. This is estimated from our data, whereby the minimum observable SdS/SS amplitude ratio is approximately 0.0065

**Fig. 8** Velocity gradients at the estimated depths of reflectors for S20RTS and SEMUCB-W1. **a** Combined bins, S20RTS[48]. **b** All bin sizes, S20RTS. **c** Combined bins, SEMUCB-W1[49]. **d** All bin sizes, SEMUCB-W1. Circles represented observed reflectors. Colours correspond to bin sizes as indicated in the key in **b**, **d**. Lines correspond to the velocity gradient from PREM[42]. Velocity gradients are calculated as the average in the bin, across a vertical distance of ±25 km, centred on the reflector. The smallest cap sizes deviate furthest from the average velocity gradient. The gradient tends towards the global average as bin size increases, as expected

mapped reflectors are laterally coherent (in depth and polarity) over thousands of kilometres, and thus witness large-scale mantle structure, and not just small-to-mid scale heterogeneity.

In eastern South America, another well-resolved 'neutral region', we observe three reflectors with alternating polarity (positive at 850, negative at 1000 and positive at 1100 km depth; Fig. 6a, c). These may have similar origins to those in the North Pacific, but a different configuration (e.g., stacked fossil slab on top of the compositional/textural change and/or complex geometrical configuration). Local accumulations of subducted oceanic or continental material may generate further regional reflectors as a consequence of heterogeneities and phase changes[25, 26, 54, 57].

Our global-scale observations provide the first detection of widespread reflectors associated with heterogeneity in the lower mantle. Significant variation in reflector geometry, depth, and polarity indicates that the underlying mechanisms arise from distinct origins in tomographically diverse domains. As reflections are most likely to occur across large-scale compositional boundaries, this study is a step towards mapping geochemical reservoirs that host subduction-related[59] and/or primordial materials[60] in the convecting mantle. Our study also provides new evidence for a potentially long-lived reservoir associated with large-scale heterogeneity in the neutral mid-mantle regions[50]. Future work is required to better characterise large-scale compositional heterogeneity in the lower mantle and orient our observations into the context of modern mineral-physics experimentation as well as geodynamic modelling. The configuration of any observed reflectors ultimately informs about the

geometry of mantle reservoirs, as well as the regionally diverse style and history of mantle flow and mixing.

## Methods

**Data and processing.** The seismic phase SS corresponds to mantle shear waves that reflect once at the Earth's surface (Fig. 1a). Underside reflections of seismic energy from deeper mantle reflectors generate precursors to SS. Interrogating SS precursors benefits from a near global coverage of mantle shear-wave structure (Fig. 1b). We have compiled a high-quality dataset of 45,634 hand-picked SS arrivals (Fig. 1c). The data are stacked into vespagrams using common mid-points for regional bins of sizes dependent on data density (various examples are shown in Fig. 2, and Supplementary Figs. 1–3) to reveal the small amplitude precursors not visible in individual seismograms.

We benefit from the recent expansion of available seismic data, meaning that this is the largest hand-picked dataset of SS precursors to date. Although our dataset spans nearly 30 years, approximately half of our data is from the past 7 years, as a result of the recent increase in seismic data coverage. Even so, data density is still poor in many areas. Fig. 1b shows the geographical coverage of the SS bounce points. Note that this does not correspond to sensitivity, however; we also require ample azimuthal and epicentral distance variation across a region to obtain slowness resolution. Supplementary Fig. 13 contains the entire dataset as a function of epicentral distance and azimuth; both show good coverage globally, but the variation in each is clear from the plots. Correspondingly, some regions therefore suffer a lack of ray paths across the full distance and azimuthal ranges, explaining why we do not retain bins in some regions with apparently sufficient data coverage. Our proportional data coverage for each domain agrees well to the global distribution of domains, and as expected, absolute data coverage increases with bin size (Supplementary Table 2; also see Supplementary Fig. 9).

We downloaded data from IRIS for every suitable event from January 1988 to April 2016. Our event criteria involve magnitudes from 6.0 to 7.0 and focal depths shallower than 30 km. We obtain data from stations in the event-receiver epicentral

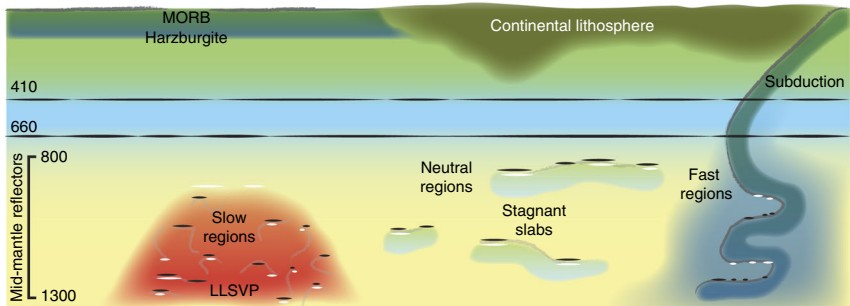

**Fig. 9** Conceptual interpretation of our observations of three distinct mantle domains. Potential sub-horizontal mid-mantle reflectors are denoted black (positive impedance) and white (negative impedance) and grouped by seismic domains: (1) Fast, cold downwelling regions (blue), with heterogeneities that are predominantly too small and variable to be resolved by SS precursors (major example region: Eastern Europe). (2) Slow, hot upwelling regions (red) (major example region: South-central Pacific). (3) Neutral regions (yellow), perhaps with compositionally or texturally distinct material (lighter yellow). Slabs may stagnate above these features, generating shallow reflections for the neutral domains (major example region: northern Pacific). Basaltic heterogeneity is denoted by black contours, with grey lines indicating heterogeneities that do not necessarily correspond to reflectors. Other known global reflectors in the mantle are indicated by black lines; e.g., at the 410 and 660 km depth

distance range from 100° to 180°. The data are first deconvolved from the receiver response to displacement, rotated to the transverse component and then filtered from 15 to 75 s for picking of individual data. We initially perform automated quality checking by removing any seismogram with a root mean square amplitude in the precursor window >0.3 of the SS signal amplitude. The data are then hand-picked by event to ensure consistency of SS waveforms and also the part of the SS that was picked. Final quality checking is performed at this stage to remove any seismograms with large amplitude arrivals in the precursor window or inconsistent SS waveforms within an event.

**Stacking**. Following picking, the data are aligned on the SS peak. For stacking, we use a relatively short period filter of 10–50 s, maintaining the original SS pick times (realigned to the position of the maxima of the SS phases). The data are then normalised according to SS amplitude. We partition the data into overlapping spherical caps based on their bounce points, to generate regional maps. The geometry is such that the centre point of a bin corresponds to the edge of each adjacent bin. Generally, even reflectors from the major 410 and 660 km discontinuities are too small to be detected in all but the highest-quality individual seismograms (Fig. 1b). Therefore, the binned data are stacked into vespagrams (Supplementary Fig. 1), which suppress incoherent noise and reveal small but coherent seismic phases.

Red and blue signals in vespagrams in Fig. 2 and Supplementary Figs. 1–3 correspond to arrivals of seismic energy. A global stack reveals the major discontinuities at 410 and 660 km depth but no global features in the mid-mantle, consistent with 1-D seismic velocity models[42, 43]. SS precursors are identified within vespagrams using theoretical arrival time and slowness with respect to SS. The cross-sections beneath the vespagrams (Fig. 2, Supplementary Fig. 1) are taken through the predicted arrival time and slowness of SS precursors with respect to SS (dashed line), calculated for PREM[42] with the TauP toolkit[61], which computes theoretical ray paths of seismic phases. Signals away from this line are not SS precursor energy. Using bootstrap resampling with 300 random resamples per stack, we estimate the 95% confidence levels (two standard deviations) of our data by calculating the standard deviation of the bootstrapped stacks. Any red-filled peaks in the cross-sections in Fig. 2 and Supplementary Figs. 1–3 have a 95% confidence level above zero and are hence significant.

**Quality checking**. After stacking, we perform quality checks for the vespagram of each bin. We discard any bins for which the 410 and 660 km discontinuities cannot be identified with certainty. We then remove stacks with significant noise in the precursor window or with poor slowness resolution. Significant noise is defined as non-precursor energy (i.e., away from the predicted arrival time and slowness) with comparable energy to that on the predicted precursor arrival and slowness line. In this case, we cannot establish whether the arrivals are actually deflected precursors or scattered noise energy. Poor slowness resolution, where energy extends across multiple slownesses, means that it is not possible to determine the true incoming slowness and hence whether the signals are SS precursors or not.

Following this quality control, we rank our remaining data by quality of the SS precursor observations, using the non-precursor noise and slowness resolution. Examples of high-quality 'A' vespagrams are shown in Fig. 2 and Supplementary Fig. 2. We define any significant peak in the precursor window along the theoretical arrival time and slowness line as an observation of a mid-mantle feature (Fig. 2a–c). We analyse vespagrams with rather large slowness ranges of −2 to +2 s/deg to confirm that the detection is indeed an SS precursor and not energy leaking from an arrival with a different slowness. Care is also taken to avoid picking potential side lobes of the 660 km precursor; we do not interpret any signals with an

estimated depth of <800 km, corresponding to 280 s before SS. No significant arrivals in this window are a negative detection (Fig. 2d; crosses in Fig. 3).

Examples of intermediate-quality 'B' and lower-quality 'C' vespagrams are included in Supplementary Fig. 3; an observation and a non-observation are shown for both quality rankings. 'B' quality data are characterised by an increase in energy away from the predicted arrival time and slowness of the precursors to result in a slightly noisier vespagram but no interference with the arrivals of interest. 'C' quality data is noisier throughout the vespagram, with non-significant energy arriving along the theoretical prediction, and less consistency in the arrivals of S410S and S660S. The importance of our statistical analysis is highlighted here, allowing us to discard energy that arrives with the expected theoretical time and slowness curve but is not significant.

**Measurements and observations**. The arrival times and amplitudes of the precursors relative to SS are used to calculate the depths and impedance contrasts of mid-mantle reflectors. We use the cross-sections taken through the vespagrams at the theoretical arrival time and slowness of the precursors relative to SS to make measurements of the arrival times and amplitudes of the precursors with respect to the SS phase. The SS waveforms are cross-correlated with both positive and negative arrivals in the precursor window. This identifies waveforms that have a similar shape to SS and are therefore likely to be SS precursors. Here we make use of the bootstrapped vespagrams to ensure only the robust SS precursors are measured. We then measure differential travel time residuals of the precursors with respect to PREM[42], using the Seismic Analysis Code (SAC)[62]. The arrival times of the SS precursors are taken at the time of the maximum amplitude of the phase. Based on the relative arrival times, we calculate estimates of the depth of the discontinuities using TauP by introducing theoretical reflectors at all depths.

The amplitude ratios of the precursors relative to SS are also measured. We use the maximum precursor signal amplitude within ±5 s of the cross-correlated arrival time. This also corresponds to the picked arrival time. The precursor/SS amplitude ratios are then corrected for the path difference of SS and of the precursors, incorporating the differing influence of geometrical spreading as well as upper mantle attenuation. This provides us with reflection coefficients. We ultimately obtain the estimated impedance contrasts by calculating reflection coefficients for theoretical mid-mantle discontinuities.

Following conversion of precursor-SS travel time residual to depth, the depths are partitioned into vertical bins of 25 km, in order to help suppress any 3-D velocity variations within the vicinity of the reflectors, as well as negate errors due to measurement uncertainties. Corrections for 3-D velocity structure (see main text) reveals that the standard deviation in depth errors is approximately 14 or 17 km, depending on the model. We thus estimate that partitioning our depth observations into 25 km radial bins should yield robust results.

Partitioning the reflectors by depth is also useful for our later interpretation of the origins of the reflectors, since the depths of a specific reflector arising from a phase change may vary laterally due to external factors, such as temperature. Performing travel time corrections for each SS bin may help to improve constraints, yet will also introduce unanticipated errors due to any discrepancies in the velocity model employed. For example, altering various travel times will influence the focussing of precursors within the vespagrams, in turn affecting their observability, and measured arrival times and amplitudes. Furthermore, the two models calculate different corrections for the data.

**Correlation to velocity domains and statistical calculations**. We characterise the reflectors according to their seismic tomography domain using clustering analysis of five different tomography models[40]. This process classifies regions into

clusters based on similar seismic properties; regions are defined as seismically fast, slow or neutral/average. The clustering was performed for five tomography models, which then generated vote maps. For plotting, we define fast or slow regions as those with three or more votes. Across each bin, we calculate the proportion of votes for each seismic domain. For bins with reflectors, we evaluate the average of the votes at the depth of the observed reflector, across the bin. For bins with no reflectors, we average the results of the cluster votes over the entire depth range of 800–1300 km at intervals of 50 km. The bin is assigned the average votes for each seismic domain, totalling five per bin. Note that all of the bins with reflectors necessarily display no reflectors at the majority of depths explored, and so our observations and statistics are heavily skewed towards and characterised by the detections rather than non-observations.

We calculate the statistical significance between two of the seismic domains, comparing the quantity of observations and their polarity. We employ a one-tail $z$-test, to obtain the probability that the observations and polarities from any two types of seismic domain are significantly different from one another. Table 1 shows the observational results for each, with significance indicated, and Supplementary Table 1 contains the calculated $p$-values.

**Travel time uncertainties and 3-D velocity corrections**. The errors in our calculated depths are estimated from the bootstrap resampling. For each observation, we generate 300 random resampled datasets and restack within ±15 s of the original reflector arrival time. The arrival time of the maximum peak in this window is used as an estimate for the reflector arrival time in each bootstrapped stack. The standard error on the mean of these 300 picks provides an estimate of the error on the arrival time of the reflector in the original vespagrams; i.e., how the arrival time would vary if data coverage differed. The mean error on all travel time measurements is 4.5 s, corresponding to an average depth error of 22 km. The errors on the picks of the arrival times are calculated by combining the sampling rate of the cross-sections through the vespagrams (0.1 s) to give a total picking error of 0.14 s.

Further uncertainties are calculated based on 3-D velocity corrections from two shear-wave velocity models: S20RTS[48] and SEMUCB-W1[49]. These are not incorporated as errors since they are not measurement uncertainties. In order to estimate the influence of 3-D velocity structure, we calculate 3-D tomography corrections for the models and calculate the average 3-D residual for each bin. For each seismogram, we use ray tracing through the models to obtain the delay time of S1000S with respect to SS. We also correct the individual data for the two shear-wave velocity models, using the theoretical delay times for S1000S with respect to SS. The data are then re-stacked into vespagrams; we recreate the four high-quality observations in Fig. 2 (Supplementary Figs. 11, 12). We select 1000 km depth as the reference since it is near to the mid-point of our depth range and our depth of interest.

**Data sensitivity**. The data are sensitive to near-horizontal reflectors and lateral variation of length scales that depend on the size of the bin. The minimum lateral size is therefore approximately 500 km for the 5° bins ranging up to 1500 km for the 15° bins. The lateral resolution tends to decrease as data density decreases; and bin sizes necessarily increase in order to obtain enough data for successful stacking. Here we band pass filter our data at periods of 15–50 s. Correspondingly the size of the SS Fresnel zone is also fairly large, on the order of 1000 km, and further complicated by its mini-max shape[44]. This can introduce errors into depth calculations, which are somewhat negated via averaging by binning and stacking the data.

In order to investigate the lateral extent of the discontinuities, we explore different cap sizes of 5°, 7.5°, 10° and 15°. There are significant discrepancies between our results for the different sizes of spherical caps, attesting to the variable length scales of heterogeneity. In the 5° cap results, several areas show detections of mid-mantle discontinuities, which vary on shorter length scales than those of larger bins. This finding indicates that lateral variation of the depth (and impedance contrast) of mid-mantle reflectors is averaged in larger bins. Other regions display observations in small bins and non-observations in large bins; indicating that some reflectors are either not large enough across larger bins to produce coherent detections or their depth varies too much across the length scale of the larger bin to stack coherently. This is corroborated by reflector properties tending towards an average as bin size increases, with differences between domains no longer significant for the largest bins. Analogously, the absence of a global mid-mantle discontinuity is not inconsistent with the widespread presence of regional reflectors.

Larger caps generally display larger signal-to-noise ratios as a result of more data in the stack and averaging over the lateral heterogeneities. However, this lateral smearing of heterogeneity becomes an issue for detecting smaller reflectors, as described above. Conversely, smaller cap sizes are too noisy in many regions and suffer from poor data coverage in some areas. To resolve this issue, we iteratively complete the map in Fig. 3 by systematically populating empty areas with increasingly large cap sizes. Although this approach generates a greater number of bins in regions with the highest data density, we prefer it as it allows us to generate higher resolution imaging where possible and provide greater global coverage than one bin size alone can provide. Maps with globally constant cap sizes are shown in Supplementary Figs. 4 (depth of discontinuity) and 6 (precursor/SS amplitude

ratio); corresponding histograms for separate bins sizes are given in Supplementary Figs. 6 and 7.

The data coverage is highly variable depending on bin size (Supplementary Fig. 14). Supplementary Fig. 8 displays the data coverage for the map of combined bins (Fig. 3), which shows the maximum geographical sensitivity of the dataset. The corresponding data coverage for each different bin size is shown in Supplementary Fig. 11. The significant variation in data coverage between bins of different sizes is a consequence of the stacking process eliminating bins with insufficient data. We also note that the combined bin coverage (Fig. 5a) appears to be poorer than the 15° bin coverage (Supplementary Fig. 9d). This is the direct result of our iterative method of completing the map; we do not incorporate larger size bins that overlap the already incorporated smaller bins by more than the bin radius, since this would result in redundant double counts for some reflectors.

The sensitivity of SS data to horizontal discontinuities is estimated using the wavelength $\lambda$ of our filtered data. Discontinuities that occur over radial length scales of $>\lambda/4$ cannot be detected, as they do not generate reflections. For our data with periods of 15–50 s, this corresponds to approximately 65 km. To calculate this length scale, we use the central value of the frequency range (23 s) with a mantle wave velocity of 6 km/s. This indicates that any reflectors we detect must arise from primarily compositional differences. Thermal gradients generally occur on vertical length scales on the order of hundreds of kilometres and thus are too gradual to be observed using SS precursors. However, their presence influences the depths of mantle discontinuities, causing shallowing or deepening of transitions. Any extremely large thermal gradients may help to generate reflectors; for example, the tops of LLSVPs may have some thermal contribution (particularly for the negative polarity reflectors), although such a gradient must occur across a vertical distance of <65 km.

**Observability of reflectors related to strength and size**. We tested the observability of SS precursors, through exploring the amplitudes of reflections generated by horizontal reflectors of varying lateral size relative to the bin and impedance contrast. For this, we used the 2.5-D spectral elements code AxiSEM[47], which generates full wavefield synthetics, incorporating attenuation and other real Earth effects. AxiSEM is selected as it allows for the incorporation of a 2-D structure; which permits us to synthesize discrete horizontal reflectors corresponding to our observations. Here we model the reflectors as regional velocity perturbations to a background model by introducing a discontinuity without a hardwired velocity jump at 1000 km.

Using PREM[42] as a background model, reflectors of varying lateral size and shear-wave velocity contrast are placed at 1000 km depth. Within the event-station geometry, they are located to be centred on the SS bounce point for the reference stacking epicentral distance of 125° (i.e., 62.5° away from the event). We explore the influence of both size and strength of the reflectors. Lateral size is varied from 5 to 25°, in increments of 5°, which corresponds to horizontal sizes of 500–2500 km at 1000 km depth in the mantle. Note that these are absolute lateral sizes of the reflectors, in contrast to the bins which are described in terms of radius. Since AxiSEM produces 2-D structures, a lateral reflector with size 25° would comprise 50% of a bin with radius 25°. The shear-wave velocity contrast is introduced as a positive perturbation to PREM, and we test values from 1% to 5% in increments of 1%. The contrast is a discontinuous step, and the velocity structure reverts back to PREM linearly over a depth of 200 km (i.e., so as not to introduce further complications from additional reflected phases). PREM attenuation is also included in our synthetic calculations.

Synthetic stations are placed every 1° from 100° to 150° event-receiver epicentral distance. Since the event location remains static, the theoretical cap size for the full epicentral distance range is therefore 25° radius (Fig. 7a), which is much larger than any bin that we employ. The different size reflectors that we introduce correspond to between 10% and 50% of this bin size, indicating the resolvability of reflectors with length scales smaller than the bins. The large epicentral distance range produces high slowness resolution. For completeness, we also stack for the smaller epicentral distance ranges that correspond to our actual bin size; we test bin sizes of 15° and 10° radius (Fig. 7b, c). Note that the slowness resolution decreases with bin size due to employing only one theoretical event for the modelling process; as a consequence, we do not model bin sizes of 5° or 7.5°.

The synthetic data are processed using the same methods as for the real data, including aligning on the major SS peak, and stacking into vespagrams. Cross-sections are taken through the synthetic vespagrams to allow for picking. The theoretical arrival times of the S1000S reflectors are calculated for PREM[42] using TauP[61], which permits for measurement of their theoretical amplitudes even when the signal cannot be identified visually in the cross-section. Using SAC[62], we finally measure the amplitude ratio of S1000S to SS (Fig. 7).

**Data availability**. Waveform data were obtained from the IRIS Data Management Center (NSF grant EAR-1063471). The processed data and measurements are available from the corresponding author upon request.

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

## Acknowledgements

L.W. is the recipient of a Discovery Early Career Research Award (project number DE170100329) funded by the Australian Government. N.C.S. and L.W. were supported by the NSF grant EAR-1361325. The research was inspired by the Cooperative Institute

for Dynamic Earth Research (CIDER) program; CIDER-II is funded as a 'Synthesis Center' by the Frontiers of Earth System Dynamics (FESD) program of NSF under grant number EAR-1135452. We thank William McDonough and James Ni for helpful discussions.

## Author contributions

L.W. compiled and analysed the data under the supervision of N.C.S. All the authors discussed the results and implications and contributed to the manuscript.

## Additional information

**Competing interests:** The authors declare no competing financial interests.

