## [Peer Review File · Nature Communications]

Reviewers' comments:

Reviewer #1 (Remarks to the Author):

Several features seen in the shallow lower mantle in regional and global tomography, including evidence for deflection of upwellings and stagnation of slabs below the transition zone, lack a clear explanation as these phenomena do not coincide in depth with known mantle material phase transitions. The authors present a large compilation of precursors to the seismic phase SS, associated with reflectors within the mantle. Identifying reflectors and investigating the association between these reflectors and mantle structures identified in tomography has the potential to yield an advance in our understanding of the origin of the dynamical behaviors seen in the mid mantle. However, I am not convinced that the correlations between properties of reflectors and seismically fast, slow, and neutral regions are robust. Figure 3a,c shows that the reflectors (or their absence) are most readily detected in the seismically slow and neutral regions of the Pacific hemisphere, implying a strong observational bias. I also do not think that the approach used to identify seismically fast, neutral and slow regions by cluster analysis of tomographic models is reasonable as it is equally likely to identify systematic errors that appear in multiple models as it is to identify robust features that appear in multiple models. I further question the value of classifying mantle domains solely based on whether they fall into three (an arbitrary number) 'bins' based solely on isotropic Vs variations. The assertion the neutral domains are associated with slab stagnation at various depths is also somewhat confusing. Are authors suggesting that lower mantle neutral domains are located beneath slabs that stagnate in the transition zone, or that the slabs themselves appear seismically neutral? I would therefore treat any correlations with a great deal of caution owing to the imperfection of the classification of fast, slow, and neutral mantle and the observational biases in the maps of scatterers. Finally, the authors infer in multiple cases (e.g. page 9, last paragraph) that the reflectors are most likely to be associated with compositional boundaries. A useful exercise that could strengthen this argument would be to place bounds on the local thermal or compositional gradient that would be necessary to explain the observed reflections.

Typo: "Their location, geographic size, depth and impedance contrast of reflectors are calculated."

I know that the authors were participants in CIDER programs in 2014 and 2016. If the work benefitted from discussions at these programs, support for CIDER through NSF should be acknowledged.

Reviewer #2 (Remarks to the Author):

Review for The Megametre Transition: Global Observations of Reflectors Map Heterogeneity in the Mid-Mantle by Waszek et al.

The authors present a large set of underside reflections, and use this data to look for mid-mantle seismic reflectors. They find a large number of reflectors, which vary both in depth, lateral consistency, and polarity. The authors then go on to correlate their observations, and the general frequency of observations, with fast, slow, and neutral regions in the mid-mantle to draw conclusions on the global nature of reflectors in the mid-mantle.

The authors have collected an impressively large data set. The observations shown in the paper are off high-quality and convincing, although I have some questions on potential artifacts due to anisotropy and reflector topography which are not addressed in the paper (see full comments below).

The observed reflectors show some interesting coherence, like a dense area of small reflectors in

the central Pacific, and at times broadly lateral coherent reflectors. However the authors proceed to make a number of claims on the global nature of these reflectors and their correlation to isotropic velocities variations. I am not convinced by their arguments of these claims with regard to a) if their data has enough global coverage to make statistically significant claims, and b) if their approach of using uneven bin distribution is structured enough to make these claims. Without these broad, sweeping claims being convincing, the paper would not be interesting to a broad audience. I hope the authors will have a chance to address and clarify my concerns, and can make their claims more statistically significant and convincing, or reword them. Lastly, I find that the title and introduction focus a lot on 1000 km, but this is not tied in the results and discussion of the paper.

Major issues and questions:

1. Can the authors show that their observations are not an artifact due to PPS and PPPS phases leaking from the radial component due to 3D effects or anisotropy? This potential artifact is shown by Zheng and Romanowicz (2012). The PPs and PPPS come in at very similar slowness and times as the underside mid-mantle reflectors across the distance range. The authors could repeat their stacks leaving out single events (to show the observations aren't biased by a single event which encounters similar upper mantle anisotropy), or show that they have sufficient distance and azimuthal coverage to not worry about 3D effects or anisotropy. Even better, they could show there is no correlated stronger arrival on the radial component.

2. The authors should state that they only image near-horizontal reflectors as they stack by mid-point. Any dipping reflector would not bounce at the mid-point and stacking would be incoherent. Are there any bins with clear arrivals from one direction and not from another, where scattered arrivals might come from outside of the bin area? Is there coverage in a dense array to apply beamforming to the scattered arrival? Please also mention somewhere the Fresnel zone sensitivity of the dominant frequency of the data and how this compares to bin sizes.

3. By eye, there appears to be a correlation between data coverage and detection of reflectors. This is of course the case when taking into account the uneven bin spaces. Can the authors show if there is a correlation in the detections using all the equal spaced bins? In other words, are all the null-detections always significant?

4. Some comments on why I don't think the overarching claims made in the paper are convincingly made:

a. 'Near absence of reflectors in seismically fast regions, correlated with dominantly sub-vertical heterogeneous slab material'

i. Supplementary text states that 27% of fast regions contain reflectors, while the table states 68%. Is the latter a typo?

ii. What percentage of the fast region is reliably sampled by the precursors? Is the sampling sufficient to make a statistically significant claim? Also when equally spaced bins are used?

b. 'reflections from the top of seismically-slow thermochemical piles beneath the Pacific;'

i. With regards to this claim, I am most worried about the uneven sampling of the bins. Only very little of the slow region (mainly central Pacific) is sampled, and this part is very densely sampled. Although I agree that the authors appear to have a highly scattered region in the Central Pacific, I do not think they can compare it to other regions using a table of percentages the way they do in the supplementary material. For the 5 degree bins, there are 20+ reflectors in the slow central Pacific patch, while for the 15 degree bin, there appear close to zero. I think the best test with regard to this, is to show the percentages given in supplementary table 1 for the different bin sizes separately, and see if any claims hold up for any bin size. A useful percentage to give is also what part of the fast and slow regions are sampled when using the different bin sizes.

c. 'abundant reflectors at variable depths in seismically neutral regions, possibly indicating a transition in composition or rock texture, and/or linked to long-term slab stagnation'

i. These reflectors are attributed to thermally equilibrated slabs in these regions in various places.

Why would fossil slabs create reflectors that aren't seen in the present-day fast regions?

5. Irrespective if the above claims are correct or not, they do not seem well represented by the cartoon in Figure 4:

a. Fast regions contain many reflectors here, including very laterally consistent in a ponding slab, and predominantly in the interior and not at the edges. Did I miss the evidence for this? Are the horizontally consistent reflectors drawn in the fast region referring to the 850 and 1000 km beneath the northern Pacific in the neutral region?

b. Overall, there appears in the cartoon a dominance of reflectors at 1000 km instead of 850 km. This is especially so in the neutral regions, where the authors appear to link these to a compositionally distinct regions. I did not see evidence for this in the results.

6. I can see how 'Megametre Transition' is an interesting term that the authors want to introduce. However, while much of the introduction pertains to discussing what is happening at 1000 km, this does not seem to play a major role in the observations of the authors (which are scattered at all depths, and somewhat focused at 850 km). The observed reflectors do not show a change in number of reflectors across 1000 km. The term 'Megametre transition' is not used in the rest of the paper, and it is not explained why the word 'transition' is used here. Something like 'Megametre Scattering Zone' would be better in the title.

7. Some of the figures mention quality categories 'Quality A, B, C.' The supplementary material mentions ranking the stacks, but no further explanation is given to what the different categories are. It is also mentioned that only high-quality stacks are shown (presumably correlated with quality A). Would the authors care to show an example of a low-quality stack (Quality C), which they are still willing to interpret ?

8. Is there any correlation between observed reflectors, their polarities, and vertical velocity gradients in tomographic models?

Other minor issues:

- Page 3, last paragraph: This review of lower mantle scattering observations could be included: Kaneshima (2016) Seismic scatterers in the mid-lower mantle.
- Last paragraph page 5 'Reflectors of small regional extent are located beneath ... eastern Europe'. I see mainly non-detections beneath Eastern Europe (but maybe a map that is not centered at the Pacific is needed).
- First paragraph page 8: 'Our observations corroborate a shallow roof of the compositionally distinct Pacific LLSVP.' Could the authors extend on this? Why does this reflect the roof of the LLSVP, and not thermochemical plume material that could be deflected at these depths? Why is Murakami et al. 2012 cited here?
- Reference 12: Title of this paper is 'Global observations...'
- Reference 20: Wrong paper?
- Reference 27: Wrong paper? (Please check all references!)
- Figure 1c: I'm not sure what this map projection is, but it does seem like the bounce points are denser near the equator than the poles. Potentially an equal area projection would help here. Or a map that shows how much data is included in each bin (and showing what areas do not have sufficient data coverage to be stacked).
- Figure 3a/c: It would be really helpful to the reader if areas with insufficient data coverage were greyed out.
- Figure 3b/d: Do these include just the observations plotted in 3a and c, or all the observations in Suppl. Fig. 3 and 4? The number of observations suggests the latter, but given some lateral correlations between the different bin sizes and observations, this would bias the distributions. In other words, are some reflectors counted four times when they appear in all different bin sizes? Maybe don't show a cumulative plot?

- Figure 4: Besides the comments on the cartoon above, did the authors mean to depict the continental LAB as a sharp boundary (as opposed to the smooth boundaries used elsewhere in the figure)?
- Suppl. Mat. Page 5, 3rd paragraph, 'the bin size of 25 km more than compensates', compared to what value? What are the discrepancies due to 3D velocity structure?
- I might just not understand the next sentence. Are we not interested in temperature variations? Does the comment about the temperature variations imply the reflectors are phase transitions? Maybe reword.
- Suppl. Mat. Page 5, 3rd paragraph, the comment about introducing unanticipated errors due to any discrepancies in the velocity model employed, seems unfair. Are the errors introduced expected to be on the same order or magnitude as the 'corrections'? Has this been tested? The velocity corrections here are only mentioned in terms of the interpretation of the depths, while velocity corrections could also lead to more coherent stacking between different azimuths and potentially more high-quality observations. I am not convinced that 3D velocity corrections are unnecessary or a bad idea here.
- Suppl. Fig. 3: This plot contains some half circles without a second observed reflector (e.g. upper left in plot d)
- Suppl. Fig. 3 and 4: The coastlines don't come through well in these plots (at least in my pdf)
- Suppl. Fig. 3 and 4: The authors say there is no relationship between depth and amplitude. I think adding four subplots of depth vs. amplitude of the observed reflectors here would be nice for the reader.
- Suppl. Fig 6: Please grey out areas where there is no resolution to detect reflectors note the vertical exaggeration factor of the various figures.

Response to Reviewer 1

Several features seen in the shallow lower mantle in regional and global tomography, including evidence for deflection of upwellings and stagnation of slabs below the transition zone, lack a clear explanation as these phenomena do not coincide in depth with known mantle material phase transitions. The authors present a large compilation of precursors to the seismic phase SS, associated with reflectors within the mantle. Identifying reflectors and investigating the association between these reflectors and mantle structures identified in tomography has the potential to yield an advance in our understanding of the origin of the dynamical behaviors seen in the mid mantle.

However, I am not convinced that the correlations between properties of reflectors and seismically fast, slow, and neutral regions are robust. Figure 3a,c shows that the reflectors (or their absence) are most readily detected in the seismically slow and neutral regions of the Pacific hemisphere, implying a strong observational bias.

We have plotted all maps centred on both 0° and 180° longitude, with equal area spacing, so as to better show the data coverage and observations on a global scale. Previously our observations were just shown in maps centred on the Pacific Ocean. However, there are considerable non-detections beneath Europe which were not clear in that projection. The projection centred on Europe now reveals the significant data coverage in the fast region here, and highlights the greater number of non-detections beneath Europe as compared to other regions.

We have also included figures showing our data coverage, including just data coverage for bins with no detections for comparison. This clearly shows that there is data coverage across fast regions as well as slow and neutral.

We also include a table presenting the geographical proportions of each domain type, and our data coverage distribution for each domain. These match closely, showing that our data coverage is representative of the distribution of domains.

I also do not think that the approach used to identify seismically fast, neutral and slow regions by cluster analysis of tomographic models is reasonable as it is equally likely to identify systematic errors that appear in multiple models as it is to identify robust features that appear in multiple models.

The cluster analysis removes the regions that are not clearly associated to fast, slow, and/or neutral seismic velocity. We consider only domains with 3 or more votes (out of 5). So, domains with fewer than 3 votes are not included in the statistics. Thus, we retain features that are robust in three out of the five models. We include the following in the Supplementary Material: “We define a domain as robust if it receives three votes or more from the five seismic tomography models”.

I further question the value of classifying mantle domains solely based on whether they fall into three (an arbitrary number) ‘bins’ based solely on isotropic V_s variations.

The use of the three domains is not arbitrary – fast refers to mostly slabs, and slow refers to plumes and feeders to the plumes. The idea is that the fast domains correspond to downwelling slabs, and the slow domains correspond to upwelling material. The neutral domains are not associated with up nor downwelling material. We have included our reasoning into the results section in which we discuss the domains (Results section, Correlation to seismic velocity domains sub-section):

“We analyse our data in the context of the domain in which reflectors are located, since these roughly correspond to the degree-2 structure of whole mantle convection also predicted by

global geodynamic models (39). Fast regions are commonly related to downwelling cold material (subducted slabs), whereas slow regions correspond to hot upwellings (plumes). Neutral domains are not correlated to either up or downwelling flow, and may be characterised in some regions by the impedance of radial flow, such as stagnation of slabs or plumes at various depths in the MTZ (16,20,45). By considering our observations in the context of average seismic velocity properties, we obtain an insight into the relationship of horizontal reflectors to mantle flow and deflection processes.”

The assertion the neutral domains are associated with slab stagnation at various depths is also somewhat confusing. Are authors suggesting that lower mantle neutral domains are located beneath slabs that stagnate in the transition zone, or that the slabs themselves appear seismically neutral?

Yes, both of those scenarios, which are both shown in the representative cartoon. The neutral mantle domains appear to be correlated with slab stagnation, however slabs do not necessarily stagnate above all neutral domains. The slab stagnation can be seen in tomographical models (e.g. Fukao & Obayashi, 2013). The downgoing slabs are observed as seismically fast, and then appear to become horizontal and stagnate at various depths in the mantle transition zone and mid-mantle. From the Discussion section:

“Possible mechanisms for these deep reflections are regional changes in rock texture or composition with depth (13), such as a transition from pyrolite to bridgmanite-enriched mantle (50,45). Our reflections could alternatively indicate a regional jump in viscosity, which has been proposed to occur at mid-mantle depths (14). Shallower reflections may arise from the top and/or bottom of a thermally-equilibrated (i.e., fossil) slab that stagnates atop the (textural or compositional) viscosity jump (45,14). Long-term stagnation can occur as slab sinking is impeded at a viscosity (or density) contrast to allow progressive slab warming that removes the negative buoyancy of the slab. Once oriented horizontally, a slab then becomes detectable by the SS precursor data.”

I would therefore treat any correlations with a great deal of caution owing to the imperfection of the classification of fast, slow, and neutral mantle and the observational biases in the maps of scatterers.

We show with the data coverage maps, and comparison between data coverage and non-observations, that data coverage does not correspond to observations. This is clearer when viewing the map with an alternate projection centred on Europe. We have performed additional statistical analysis for the domains for each bin sizes (Tables 1 and S1). In our discussion of the results, we have noted that for fast and slow domains, the reflectors predominantly occur near to the domain edges, in order to focus on the unexplained observations from within the neutral domains. This takes into account the observations which may fall into either the fast/slow or neutral regions, negating some of the classification problems near to the edges.

Finally, the authors infer in multiple cases (e.g. page 9, last paragraph) that the reflectors are most likely to be associated with compositional boundaries. A useful exercise that could strengthen this argument would be to place bounds on the local thermal or compositional gradient that would be necessary to explain the observed reflections.

We have estimated the required gradient via various methods.

1. A mathematical calculation from the wavelength of our data provides a rough estimate of the sensitivity of the data as a maximum radial gradient of 80 km. This is included in the Data Sensitivity subsection of the Methods sections as follows:

“The sensitivity of SS data to horizontal discontinuities is estimated using the wavelength λ of our filtered data. Discontinuities which occur over radial length scales of more than $\lambda/4$ cannot be detected. For our data with periods of 15 – 50 s, this corresponds to approximately 80 km. To calculate this length scale, we use the median value of the frequency range with a mantle wave velocity of 3.5 km s⁻¹. This indicates that any reflectors we detect must arise from compositional differences. Thermal gradients occur on vertical length scales on the order of hundreds of kilometres, and thus are too gradual to observe using SS precursors. However, their presence influences the depths of mantle discontinuities, causing shallowing or deepening of transitions.”

2. We have used the 2.5-D spectral elements code Axisem to explore the influence of size and strength of reflector on SdS/SS amplitude ratios. The amplitude ratio (and hence observability of the reflectors) is presented as a function of these parameters in Figure S15, for various bin sizes. This provides a lower bound on the shear wave velocity contrast required by the reflectors. The results of this are fully described in the subsection “Observability of reflectors: strength and size” in the Supplementary Material. The figure provides estimates of shear wave velocity contrasts which are detectable in vespagrams of high and average quality. For average quality vespagrams, a shear wave velocity contrast of about 2% is detectable for the reflector which has a comparable size to that of the bin. As the reflector size becomes much less than the bin size, the minimum velocity contrast observable increases.
3. We have calculated the estimated shear wave velocity gradient across the reflectors from the shear wave velocity models S20RTS (Ritsema et al., 1999) and SEMUCB-W1 (French and Romanowicz, 2014). We find that there is no relationship to observed reflector and velocity gradient (Figure S16), indicating that the reflectors probably result from variation on shorter length scales than may be resolved by the shear wave tomography models.

Typo: “Their location, geographic size, depth and impedance contrast of reflectors are calculated.”

Corrected to “We measure the location, geographic size, depth, and impedance contrast of the reflectors in the mid-mantle.”

I know that the authors were participants in CIDER programs in 2014 and 2016. If the work benefitted from discussions at these programs, support for CIDER through NSF should be acknowledged.

We have added the acknowledgement to CIDER.

Response to Reviewer 2

The authors present a large set of underside reflections, and use this data to look for mid-mantle seismic reflectors. They find a large number of reflectors, which vary both in depth, lateral consistency, and polarity. The authors then go on to correlate their observations, and the general frequency of observations, with fast, slow, and neutral regions in the mid-mantle to draw conclusions on the global nature of reflectors in the mid-mantle.

The authors have collected an impressively large data set. The observations shown in the paper are off high-quality and convincing, although I have some questions on potential artefacts due to anisotropy and reflector topography which are not addressed in the paper (see full comments below).

The observed reflectors show some interesting coherence, like a dense area of small reflectors in the central Pacific, and at times broadly lateral coherent reflectors. However, the authors proceed to make a number of claims on the global nature of these reflectors and their correlation to isotropic velocities variations. I am not convinced by their arguments of these claims with regard to a) if their data has enough global coverage to make statistically significant claims, and b) if their approach of using uneven bin distribution is structured enough to make these claims. Without these broad, sweeping claims being convincing, the paper would not be interesting to a broad audience. I hope the authors will have a chance to address and clarify my concerns, and can make their claims more statistically significant and convincing, or reword them.

Lastly, I find that the title and introduction focus a lot on 1000 km, but this is not tied in the results and discussion of the paper.

Major issues and questions:

1. Can the authors show that their observations are not an artefact due to PPS and PPPS phases leaking from the radial component due to 3D effects or anisotropy? This potential artefact is shown by Zheng and Romanowicz (2012). The PPs and PPPS come in at very similar slowness and times as the underside mid-mantle reflectors across the distance range. The authors could repeat their stacks leaving out single events (to show the observations aren't biased by a single event which encounters similar upper mantle anisotropy), or show that they have sufficient distance and azimuthal coverage to not worry about 3D effects or anisotropy. Even better, they could show there is no correlated stronger arrival on the radial component.

For our reference stacking distance of 125°, PPS and PPPS are estimated to arrive with similar travel times and slownesses to the mid-mantle reflectors. Crucially, either the arrival time or the slowness differ enough to distinguish the signals within a vespagram with good slowness resolution. The table below calculates the theoretical travel time and slowness of the potentially interfering phases with respect to SS:

Phase	Relative travel time (s)	Relative slowness (s/deg)
PPS	357.36	-2.648
S1200S	335.49	-1.23
PPS	331.06	-2.173
S1150S	326.6	-1.155
S1100S	317.54	-1.082
S1050S	308.3	-1.007
S1000S	298.88	-0.934

S950S	289.27	-0.864
PPPS	282.25	-0.397
S900S	279.48	-0.799
PPPS	277.14	-1.179
PPPS	276.74	-1.021
S850S	269.51	-0.729
S800S	259.34	-0.662

The table therefore shows that only energy significantly redirected by 3D mantle structure or anisotropy will affect our data. Thus, we firstly provide the distance and azimuthal distribution of our bins. We have added a plot showing the distance and azimuthal coverage to the Supplementary Material (Figure S1). This shows the good distribution across epicentral distances. Azimuthal distribution is fairly good on a global scale, showing the future potential for investigations into the variation of discontinuities as a function of distance.

The observation by Zheng and Romanowicz found that the leaking component results in the apparent S660S precursor becoming anomalously large. Our quality checking procedure means that do not retain any such seismograms with anomalous S660S or S410S precursors. Also we do not make any measurements of precursors near to the S660S arrival; the shallowest depth we detect is 800 km. Re-stacking all of our data for the radial component is beyond the scope of this study, and represents the next stage for future work.

We test the influence of inaccurate station parameters using synthetic data. We generate PREM synthetics for a bin size of 15°. After rotating the N and E components to the great circle path to obtain the transverse component, we then rotate it again by angles of 1° through 5° to explore the influence of leaking from the radial component. These results are presented in Figure 1 (at the end of this document). Figure 1a is the original synthetics, with b-e showing increasingly incorrect rotation angles.

At rotation angles smaller than about 2°, there is no issue with energy arriving from the radial component and interfering. Energy leakage starts to become clear at 3° of rotation, arriving at around -275 s with respect to the SS phase. At this point, the waveform of the S660S signal also becomes distorted, with significant distortion at the largest tested rotation angle of 5°. Consequently, any such vespagrams would be discarded, since one of our initial quality checks is the waveforms of S410S and S660S. Furthermore, we do not use any precursors with arrival times of less than 280 s with respect to SS, in case of misinterpreting sidelobes from S660S. Thus, these signals would not be picked and interpreted as precursors anyway. For larger rotation angles, energy from the radial component is apparent at around -320 s, however this is clearly away from the theoretical slowness line of SS precursors and would not be mistaken for an SS precursor during our quality processing.

2. The authors should state that they only image near-horizontal reflectors as they stack by mid-point. Any dipping reflector would not bounce at the mid-point and stacking would be incoherent. Are there any bins with clear arrivals from one direction and not from another, where scattered arrivals might come from outside of the bin area? Is there coverage in a dense array to apply beamforming to the scattered arrival? Please also mention somewhere the Fresnel zone sensitivity of the dominant frequency of the data and how this compares to bin sizes.

To the final paragraph of the Introduction section we included the following clarification sentence:

“We demonstrate that our dataset is sensitive to near-horizontal reflectors with length scales on the order of 500 to 1,500 km.”

We also incorporated the following into our discussion of the bins with non-detections, in the General Observations sub-section of the Results section:

“Due to the mid-point stacking technique, any reflectors which are not oriented (sub-)horizontally, such as dipping structures, will not stack coherently.”

Our bins do not indicate the direction of energy, since they stack over all azimuthal ranges.

This is a benefit of our study, since it averages over the 3D regional structure in the mantle.

The question of anisotropy in the mid-mantle will be addressed in a future study.

We have included a discussion of Fresnel zone sensitivity to the Data Sensitivity sub-section of the Methods section:

“The data are sensitive to near-horizontal reflectors, and lateral variation of length scales which depend on the size of the bin. The minimum lateral size is therefore approximately 500 km for the 5° bins ranging up to 1500 km for the 15° bins. The lateral resolution is larger in regions where data density is lower, and bins sizes are necessarily greater in order to obtain enough data for successful stacking. Here we band pass filter our data at periods of 15 – 50 s. Correspondingly the size of the SS Fresnel zone is also fairly large, on the order of 1000 km, and further complicated by its mini-max shape (57,58). This can introduce errors into depth calculations, which are somewhat negated via averaging by binning and stacking the data.”

3. By eye, there appears to be a correlation between data coverage and detection of reflectors. This is of course the case when taking into account the uneven bin spaces. Can the authors show if there is a correlation in the detections using all the equal spaced bins? In other words, are all the null-detections always significant?

The figures showing data coverage also incorporates the bins which display no detections, shown as crosses. In Figure 3, the absence of a symbol does not correspond to no reflector; instead, it corresponds to data for which we do not interpret due to poor coverage or quality.

To address the reviewer’s question, we have plotted maps of the non-detections for all bin sizes for comparison. We have also plotted the data coverage for all of the bins, as well as the data coverage for just the non-observations only.

This resolves the query regarding correlation between data coverage and detection of reflectors: the maps clearly show that data coverage does not correspond to the observation of a reflector. Thus, the null detections are always significant – they image regions which do not have a significant near-horizontal reflector on the length scale of the bin.

4. Some comments on why I don't think the overarching claims made in the paper are convincingly made:

a. 'Near absence of reflectors in seismically fast regions, correlated with dominantly sub-vertical heterogeneous slab material'

i. Supplementary text states that 27% of fast regions contain reflectors, while the table states 68%. Is the latter a typo?

This is not a typo. The percentage arises from the use of the bins at every depth vs taking an average over the depths. I.e. a non-observation in a bin corresponds to multiple non-observations at various depths. In our revision we have taken an average over the entire bin for the non-observations, and used this as one single non-observations. We have taken care to highlight that this is how we calculate the statistics in the main manuscript and in the description of Table 1:

“For bins with no reflector, the average cluster votes are taken across the entire depth range, for the centre location of the bin. Note that many bins with observations have just one detection, and thus the remainder of the bin may also be considered to display an absence of reflector; this is not accounted for in the statistics.”

The table now shows that 71% of bins with centres on fast domains for the combined bins

contains observations. Treating the non-observations as a non-observation every 50 km depth from 800 to 1350 km, we calculate a value of 27%. This is not included in the manuscript, so as to prevent unnecessary confusion for the readers.

ii. What percentage of the fast region is reliably sampled by the precursors? Is the sampling sufficient to make a statistically significant claim? Also when equally spaced bins are used? We have calculated the percentage of fast regions sampled by the precursors. This is included in the Supplementary Material in Table S1. We also calculate the relative proportions of domain types globally, and sampled by our data set. This shows that our data coverage closely matches the actual distribution of domains. We added the following to the manuscript to highlight this:

“Our proportional data coverage for each domain agrees well to the global distribution of domains, and as expected, absolute data coverage increases with bin size (Table S1; also see Figure S10).”

b. 'reflections from the top of seismically-slow thermochemical piles beneath the Pacific;'

i. With regards to this claim, I am most worried about the uneven sampling of the bins. Only very little of the slow region (mainly central Pacific) is sampled, and this part is very densely sampled.

We have now added data coverage to Figure 3, which clearly shows that the slow region in the Pacific is well-sampled by our data (areas with no hatching in the background represents areas with data coverage).

Although I agree that the authors appear to have a highly scattered region in the Central Pacific, I do not think they can compare it to other regions using a table of percentages the way they do in the supplementary material. For the 5 degree bins, there are 20+ reflectors in the slow central Pacific patch, while for the 15 degree bin, there appear close to zero. I think the best test with regard to this, is to show the percentages given in supplementary table 1 for the different bin sizes separately, and see if any claims hold up for any bin size. A useful percentage to give is also what part of the fast and slow regions are sampled when using the different bin sizes.

We have added the results for the different bin sizes, and used these to compare rather than the combined bin sizes. These results show that the percentage of bins with observations varies with length scale for the slow domains. This agrees very well with the reviewer's comment, that although there are abundant reflectors for the 5° bins, there are much fewer for the larger bins, highlighting the short length scale of variation in the slow domains. This has been added into the results section describing the observations in different domains.

We have included a table which shows the percentage of fast and slow regions sampled for each bin size (Table S1). It contains the global proportion of each domain type, the percentage of the domain type that our data samples, and the relative proportion of each domain type sampled. The proportion of our data coverage and the domain types match closely, and the table shows the increase in data coverage as bin size increases.

c. 'abundant reflectors at variable depths in seismically neutral regions, possibly indicating a transition in composition or rock texture, and/or linked to long-term slab stagnation'

i. These reflectors are attributed to thermally equilibrated slabs in these regions in various places. Why would fossil slabs create reflectors that aren't seen in the present-day fast regions?

The present day fast regions are downwelling slabs which are not oriented horizontally. The reflectors that we detect must correspond to laterally continuous horizontal features, and thus

correspond to slabs which are stagnant (and thus, by definition, are mostly horizontal). We added the following to the discussion section, in the neutral region paragraph:

“Long-term stagnation can occur as slab sinking is impeded at a viscosity (or density) contrast to allow progressive slab warming that removes the negative buoyancy of the slab. Once oriented horizontally, a slab then becomes detectable by the SS precursor data.”

5. Irrespective if the above claims are correct or not, they do not seem well represented by the cartoon in Figure 4:

a. Fast regions contain many reflectors here, including very laterally consistent in a ponding slab, and predominantly in the interior and not at the edges. Did I miss the evidence for this? The fast regions with multiple reflectors are supposed to represent the concept that although there is heterogeneous material here, it does not stack coherently in the SS data bins. In other words, the reflectors are generally too small and inconsistent. This is now explained in our comparison of bins sizes; i.e. that some reflectors cannot be detected in the larger bin sizes. We have also tested the influence of reflector size and strength on observability within the bins, and show that weak reflectors cannot be detected if they are much smaller than the bin size (Supplementary Material subsection “Observability of reflectors: size and strength”, Figure S15).

We also incorporate this explanation to the figure captions as follows:

“Fast, cold downwelling regions (blue), with heterogeneities that are predominantly too small and variable to be resolved by SS precursors”

We also add some discussion to the Discussion section regarding the fast regions, as follows:

“Our modelling shows that only reflectors on length scales similar to that of the smaller bin sizes may be resolved (Figure S15). Mid-mantle reflectors in these regions are consistent with small-scale heterogeneity, as may be expected from the vast range in composition of subducted material.”

Are the horizontally consistent reflectors drawn in the fast region referring to the 850 and 1000 km beneath the northern Pacific in the neutral region?

No. The reflectors drawn in the fast region refer to the reflectors we detect in the fast regions. The northern Pacific is represented by the neutral region, with the stagnant slab oriented horizontally above. We have added our best example regions to the caption of the cartoon in order to describe this more clearly, as follows:

“1) Fast, cold downwelling regions (blue), with heterogeneities that are predominantly too small and variable to be resolved by SS precursors (major example region: Eastern Europe). 2) Slow, hot upwelling regions (red) (major example region: south-central Pacific). 3) Neutral regions (yellow), perhaps with compositionally or texturally distinct material (lighter yellow). Slabs may stagnate above these features, generating shallow reflections for the neutral domains (major example region: northern Pacific).”

b. Overall, there appears in the cartoon a dominance of reflectors at 1000 km instead of 850 km. This is especially so in the neutral regions, where the authors appear to link these to compositionally distinct regions. I did not see evidence for this in the results.

The neutral region beneath the north Pacific is dominated by a reflector at 1000 km (green circles). The neutral region also incorporates shallower reflectors in the stagnant slab. We have updated the caption to clarify that the stagnant slab should be linked to the neutral regions, rather than a fast region, as follows:

“). 3) Neutral regions (yellow), perhaps with compositionally or texturally distinct material (lighter yellow). Slabs may stagnate above these features, generating shallow reflections for the neutral domains (major example region: northern Pacific).”

6. I can see how 'Megametre Transition' is an interesting term that the authors want to introduce. However, while much of the introduction pertains to discussing what is happening at 1000 km, this does not seem to play a major role in the observations of the authors (which are scattered at all depths, and somewhat focused at 850 km). The observed reflectors do not show a change in number of reflectors across 1000 km. The term 'Megametre transition' is not used in the rest of the paper, and it is not explained why the word 'transition' is used here. Something like 'Megametre Scattering Zone' would be better in the title.

We have changed the title and manuscript to remove “Megametre”, and use mid-mantle instead. The title is now:

“Global observations of reflectors in the mid-mantle: implications for mantle structure and dynamics”

7. Some of the figures mention quality categories 'Quality A, B, C.' The supplementary material mentions ranking the stacks, but no further explanation is given to what the different categories are. It is also mentioned that only high-quality stacks are shown (presumably correlated with quality A). Would the authors care to show an example of a low-quality stack (Quality C), which they are still willing to interpret?

We have included examples of quality B and C vespagrams, showing an example of an observation and a non-observation, in the supplementary material (Figure S3). We add some discussion of these vespagrams to the Methods section, Quality Checking subsection:

“Examples of intermediate quality “B” and lower quality “C” vespagrams are included in Figure S4; an observation and a non-observation are shown for both quality rankings. “B” quality data is characterised by an increase in energy away from the predicted arrival time and slowness of the precursors, to result in a slightly noisier vespagram but no interference with the arrivals of interest. “C” quality data is noisier throughout the vespagram, with non-significant energy arriving along the theoretical prediction, and less consistency in the arrivals of S410S and S660S. The importance of our statistical analysis is highlighted here, allowing us to discard energy which arrives with the expected theoretical time and slowness curve, but is not significant.

8. Is there any correlation between observed reflectors, their polarities, and vertical velocity gradients in tomographic models?

We have now included a section in the Supplementary Material exploring the relationship between the reflector depths and polarities and the vertical velocity gradients. For this we use the global shear wave velocity models S20RTS and SEMUCB-W1 for the combined bins and for the different bin sizes separately (all shown in Figure S16).

For each set of observed reflectors, we obtain the average shear wave velocity gradient in the vertical region within ± 25 km of the reflector depth. We incorporate the size of the bin into this value, by calculating the average lateral gradient across the full bin size.

Other minor issues:

- Page 3, last paragraph: This review of lower mantle scattering observations could be included: Kaneshima (2016) Seismic scatterers in the mid-lower mantle.

This reference has now been included.

- Last paragraph page 5 'Reflectors of small regional extent are located beneath ... eastern Europe'. I see mainly non-detections beneath Eastern Europe (but maybe a map that is not centered at the Pacific is needed).

For every figure with a map, we have added a further subplot showing a map centred on the

UK to complement those centred on the Pacific, to all plots in the manuscript and supplement. We have also changed the map projection to the Mollweide projection, which is of equal area. This allows the reader to see the distribution of observations and non-observations, as well as data coverage, much more clearly.

- First paragraph page 8: 'Our observations corroborate a shallow roof of the compositionally distinct Pacific LLSVP.' Could the authors extend on this? Why does this reflect the roof of the LLSVP, and not thermochemical plume material that could be deflected at these depths? Why is Murakami et al. 2012 cited here?

We have rephrased the sentence to include the possible origin of our observations as follows: "Streaks of basalt/harzburgite would produce alternating bands of elevated and lowered seismic velocity and density contrasts, similar to the observed impedance contrasts within the data. This interpretation implies that the numerous reflectors within the seismically slow region map the shallow roof of a compositionally-distinct Pacific LLSVP (37,50) (Figure 6)." Murakami et al., 2012, is cited to refer to the finding that the mantle is chemically stratified resulting in layered mantle convection.

- Reference 12: Title of this paper is 'Global observations...'
This has now been corrected.

- Reference 20: Wrong paper?

It is not clear why the reviewer has suggested that this is the wrong paper. This is the paper we intended to cite at this point in the text. The reference is:

22. Tosi, N., Yuen, D. & Cadek, O. Dynamical consequences in the lower mantle with the post-perovskite phase change and strongly depth-dependent thermodynamic and transport properties. *Earth. Planet. Sci. Lett.*, **298**, 229-243 (2010).

and referred to in the following context:

"While deflections of mantle flow near the 660 have been related to the effects of a major phase transition (21,22), those near 1,000 km have instead been ascribed to the presence of a viscosity jump (14,23), radial change(s) in mantle composition (13), and/or regional mineral phase changes (24,25)."

- Reference 27: Wrong paper? (Please check all references!)

It is not clear why the reviewer suggests that this is the wrong paper. This is the paper we intended to cite at this point in the text. The reference is:

32. Jenkins, J., Cottar, S., White, R. & Deuss, A. Depressed mantle discontinuities beneath Iceland: evidence of a garnet controlled 660 km discontinuity? *Earth. Planet. Sci. Lett.*, **433**, 159-168 (2015).

and referred to in the following context:

"Studies also find evidence for reflectors in regions of upwelling, such as the Hawaiian and Icelandic hotspots (18,31-33)."

- Figure 1c: I'm not sure what this map projection is, but it does seem like the bounce points are denser near the equator than the poles. Potentially an equal area projection would help here. Or a map that shows how much data is included in each bin (and showing what areas do not have sufficient data coverage to be stacked).

The bounce points are equally spaced on the globe, so the difference in density is indeed an artefact of the map projection as the reviewer notes. We have changed all of the map projections to the Mollweide projection, which is equal area.

We have also added to the Supplement a figure which shows the number of data per bin as a

function of a circle size (Figure S14). The corresponding maps for the combined bin plot are shown in Figure 5c. Figure S10 shows the areas which do not have enough high quality data to be picked (the non-shaded areas).

- Figure 3a/c: It would be really helpful to the reader if areas with insufficient data coverage were greyed out.

We have greyed out the areas with insufficient data coverage. We have also included figures showing data coverage and distribution to the main manuscript (Figure 5), which includes for the non-detections. We have also added figures showing data coverage for all bins sizes separately (Figure S10), as well as the corresponding plots for the non-detections (Figures S13), as well as a figure showing data distribution (Figure S14).

- Figure 3b/d: Do these include just the observations plotted in 3a and c, or all the observations in Suppl. Fig. 3 and 4? The number of observations suggests the latter, but given some lateral correlations between the different bin sizes and observations, this would bias the distributions. In other words, are some reflectors are counted four times when they appear in all different bin sizes? Maybe don't show a cumulative plot?

The observations were cumulative (as stated in the caption), i.e. they show all of the observations for all of the cap sizes which corresponds to the figures in the Supplementary Material. We have now changed this to show just the observations from the combined bin plot (Figure 4). We have also generated depth and amplitude ratio histograms for each bin size separately, to allow for comparison. These have been added to the Supplementary Material (Figures S7 and S8).

- Figure 4: Besides the comments on the cartoon above, did the authors mean to depict the continental LAB as a sharp boundary (as opposed to the smooth boundaries used elsewhere in the figure)?

No. The sharp boundaries which correspond to reflectors are marked by the black and white lines, and not by colour contrasts. We have edited the cartoon to blur out this sharp boundary.

- Suppl. Mat. Page 5, 3rd paragraph, 'the bin size of 25 km more than compensates', compared to what value? What are the discrepancies due to 3D velocity structure?

We calculated the discrepancies due to 3D velocity structure using ray tracing through S20RTS and SEMUCB-W1, for S1000S with respect to SS. We calculate a maximum delay time of 7.4 s in one bin, and a standard deviation on the delay times of 2.8 s. The latter corresponds to a vertical distance of approximately 15 km. Thus, by binning our data into 25 km bins, we more than compensate for this discrepancy. We have added an explanation of this to the Methods section, subsection "Travel time uncertainties and 3-D velocity corrections".

We explore the effects of 3-D velocity structure further by correcting our data for the two seismic tomography models, and restacking. The methods are discussed in the Supplementary Material, and corrected vespagrams are shown in Figure S11 and S12.

- I might just not understand the next sentence. Are we not interested in temperature variations? Does the comment about the temperature variations imply the reflectors are phase transitions? Maybe reword.

To answer the two questions: yes, we are interested in all variations, and no, it does not imply that the reflectors are phase transitions. However as stated elsewhere, solely thermal variations would result in transitions across radial length scales greater than can be detected by our data. Thus, we have rephrased the sentence to focus on our interpretation of the

reflectors detected here as phase transitions as follows:

“Partitioning the reflectors by depth is also useful for our later interpretation of the origins of the reflectors, since the depths of a specific reflector arising from a phase change may vary laterally due to external factors such as temperature.”

- Suppl. Mat. Page 5, 3rd paragraph, the comment about introducing unanticipated errors due to any discrepancies in the velocity model employed, seems unfair. Are the errors introduced expected to be on the same order or magnitude as the 'corrections'? Has this been tested? The velocity corrections here are only mentioned in terms of the interpretation of the depths, while velocity corrections could also lead to more coherent stacking between different azimuths and potentially more high-quality observations. I am not convinced that 3D velocity corrections are unnecessary or a bad idea here.

Our point was that the corrections may also correspond to errors if the velocity structure is not accurate, and these errors are unknown. This section has been rephrased.

We have now calculated stacks for data corrected using two 3-D S-wave mantle velocity models (S20RTS and SEMUCB). The ray paths were corrected individually before stacking, using the delay times of S1000S with respect to SS. The methods and results of this are presented primarily in the Supplemental Material, with some discussion in the main text. The two models correct the data differently, and some evidence of defocussing the arrivals is clear. We therefore retain our uncorrected data for analysis.

- Suppl. Fig. 3: This plot contains some half circles without a second observed reflector (e.g. upper left in plot d)

These have been fixed (also the corresponding map for amplitude ratios).

- Suppl. Fig. 3 and 4: The coastlines don't come through well in these plots (at least in my pdf)

The coastlines have been changed from grey to black.

- Suppl. Fig. 3 and 4: The authors say there is no relationship between depth and amplitude. I think adding four subplots of depth vs. amplitude of the observed reflectors here would be nice for the reader.

We have added four subplots of amplitude ratio as a function of reflector depth to the Supplementary Material (Figure S9). There is no correlation. We refer to this at the relevant point in the main text.

- Suppl. Fig 6: Please grey out areas where there is no resolution to detect reflectors note the vertical exaggeration factor of the various figures.

We have greyed out areas with no data coverage on both the map and the depth slices. We have included the following in the figure caption:

“Note the vertical exaggeration of the depth slices.”.

N.B. This figure is now shown in the main paper.

Figure 1. Vespagrams from synthetics generated using AxiSEM for PREM. The N and E components are firstly rotated to the great circle path to generate the transverse component. We then rotate the component again by an addition 1 though 5°, to simulate the effect of leaking energy from the radial component.
a. Great circle path. **b.** 1° rotation. **c.** 2° rotation. **d.** 3° rotation. **e.** 4° rotation. **f.** 5° rotation. **g.** 6° rotation.

Reviewers' comments:

Reviewer #1 (Remarks to the Author):

The manuscript has improved significantly since the original submission and I very much appreciate the time and effort that the authors have invested in addressing the comments from both reviewers. The manuscript should be suitable for publication once the major point is addressed.

Major point:

1. Both reviewers in the original review indicated that the correlation between reflectors and mantle domains was not well-established. To address this, the authors addressed data coverage, shown in Supplementary Table S1. This table shows the data coverage for each domain as well as the 'proportional coverage', which measures the degree to which the data coverage mirrors the distribution of fast, neutral, and slow regions in the clustered tomography models.

a. This seems a reasonable start, but stops short of performing a test for statistical significance. It remains unclear whether the correlation between fast/neutral/slow mantle domains and precursors is robust.

b. An alternative explanation for the lack of precursors in fast regions might be that the (presumably compositionally distinct) reflectors do not have as large an impedance contrast relative to their surroundings in these regions, making detection more difficult.

Minor points:

2. The authors argue for a compositional origin of the reflectors, which is certainly a reasonable possibility. However it is never established in the paper that a sufficiently large thermal gradient is impossible. Some types of reflectors, particularly the tops of the LLVPs, would have both thermal and chemical contributions.

3. As pointed out by reviewer #2, the discussion places significant emphasis on processes observed to occur around 1000 km depth. This is somewhat at odds with the depth histogram of reflectors (Figure 4) which shows a peak around 875 km. This is a similar depth to where Boschi and Becker 2011 find that introducing decorrelation in tomographic models improves model fit to traveltimes.

Reviewer #2:

I appreciate a number of the clarifications and a more toned down version of this paper. However, I still have a number of issues and questions that either haven't been clarified sufficiently (sometimes due to misunderstandings), or that have arisen from this new version. Specifically now that the tables of observations across the caps and coverage of the domains are added, it allowed me to better assess the statistical significance of the claims made (which unfortunately do not appear significant). I do not think the paper should be accepted in the current state, I encourage the authors to either do more convincing statistical analysis of the claims they make, or re-asses the message they want to give based on their observations.

- The authors claim to do 'statistical analysis' on their results, although this largely consists of giving a lot of percentages to the reader. The table is somewhat overwhelming to the reader (I do appreciate that I asked for analysis of the different bins in the first review). It would be really helpful if the authors put the values in bold, which are statistically significant from the rest? E.g. in the first line 19/21 true observations in the slow domain is statistically significant from the others, while in the 10 degree caps, the observation of 7/11 in the slow domain is not statistically significant from the other domains. Equally, in the first line the 71% of observations in the fast domain is not statistically significant from the neutral domain, and should not be sold as such in the paper. I am afraid, very little in this table stands out as statistically different.

Additionally, one could account how much is sampled on the global scale. E.g. in the 7.5 degree bin, the 4 (100%) positive observations, might seem significant, but from the supplementary material I gather these 4 observations only sample 8% of the global domain, making it not statically significant claim as the samples size is small.

Related to this point, the discussion starts off if the following sentence: *'The observed mid-mantle reflectors exhibit no geographic relationship to surface features; instead they correlate to mid-mantle structure as imaged by seismic tomography.'* In the previous chapter, the author has just shown that they do not correlate to seismically imaged vertical gradients, so I presume the authors mean there is correlation with the defined seismic domains here. In the table below, I show that when adding up all the domains, there is no significance difference of scattering within the domains. The number of reflectors observed in each domain, as well as the number of positive reflectors in each domain, correlate with the amount of bins assigned to that domain, showing now significant differences between the domains.

This table shows that when adding up all the observations across the caps there are no significant difference between the domains. The number of observations in each domain reflects the proportional data coverage.

All caps combined	Fast	Neutral	Slow
range proportional data coverage across the different caps	19-24%	68-70%	7-12%
Observations	63	20	30
% of total observations	21.4%	68.4%	10.2%
Positive	47	144	18

observations			
% or positive observations	22.4%	68.9%	8.6%
non-detections	29	99	13
% of non-detections	20.5%	70.2%	9.2%

If the authors want to build their claims on statistics, I think they should do it properly. If they want to make sweeping, qualitative statements (like is quite common in the field), don't claim to do a robust analysis and move away from words like 'correlate' and 'statistical analysis' (although I wouldn't suggest going this route). In light of broad statements, I think the observations in this paper show, dense observations of horizontal reflectors beneath the central Pacific (in the 5 degree bins), and observations of broad reflectors around 850 and 1000 km.

- The authors show a synthetic test of potential rotation of the components at the station. In this test they show energy leakage at a time and slowness, which they would interpret in their stacks as coming from a mid-mantle reflector. It is unclear to me what phase from the radial is leaking in here and the authors do not mention this (the timing and slowness of the observed arrival on the black line of predicted reflectors does not seem to correspond with the predictions in the table provided in the rebuttal). I am not so much worried about consistent rotation of the horizontal components at all the stations used, but a bias of the upper mantle anisotropy (which can be consistent over large regions and many stations) could cause leakage between the components. The authors claim to throw out similar slowness stacks due to distortion on the S660S phase, however I don't find the rotated S660S arrival in the synthetic more distorted than in the stacks shown in Figures 2a and 2b. I understand that stacking the radial component for all the bins and analysing them is beyond the scope of this research. However, I don't think it is too much to ask to stack the radial component for one good quality stack with an identified arrival and show that there is no interfering energy on the radial component.

- From the rebuttal to point 4.a, I gather that the authors switched readily between analyses the observations of the lateral bins (e.g. to determine the domain, and fill in the table), as well as a function of volume (e.g. when drawing the conclusion that there is a lack of scatterers in the fast domain). This is very confusing for the reader now, as the latter conclusion is not clear from the table or figures. Why do the authors not do all their analysis in volume space, and also let the domains vary as a function of depth? (Maybe some of the analyses would become statistically interesting in volumes space).

- The point 4.b.i I made in the first review, might have been misunderstood, as the authors explain in the rebuttal that the Central Pacific is well covered by the data. I totally agree it is well-sampled. With my point, I meant that it was over-sampled in the combined caps approach, where this area is reflected by dense grid of degree 5 caps. From table 1, I now gather that this claim of lots of observations in the slow domain is entirely biased by the 5 degree caps, which are densely sampling the mid-Pacific in the Combined caps approach. In none of the other degree caps does the slow area stand out as significantly different. In the light of the discussion that each of the various degree caps has sensitivity to different size of reflectors, the combined approach seems to not add much value.

- Figure 6 is a lot for the reader to take in, and the colorbar is given limited explanation. Now that the authors have clarified that they define their domains at 3 votes, would it not be easier to just plot the domains in the background (and one color across the whole depth range, if the authors assign a domain to each bin), instead of the full triangle of colors. This can also clarify what region is not counted as one of the domains (do any of the caps fall within a boundary region where votes don't agree? From the table in the supplementary material I gather all the domains add up to 100%. Or are observations in regions with fewer than 3 votes for one domain discarded?).

- The first conclusion in the abstract states '1) near absence of reflectors in seismically fast regions, likely related to dominantly sub-vertical heterogeneous slab material; '. I've already mentioned that I don't first half of the claim is not statistically significant, and I don't see how the second half follows on the first. Maybe state '1) near absence of broad reflectors in seismically fast regions, although previously observed small-scale sub-vertical heterogeneous slab material would not be imaged by our dataset; '. Related to this, it might be helpful to give the length-scale of potential observations in the abstract, so the reader know what type of reflectors are being referred to.

- It is also very confusing for the reader that in the cartoon that one blue bit is fast domain, and the other is neutral domain (and it appears both reviewers were confused by this). Maybe fossilized, neutralized, slab material should not be drawn as blue or connected to the surface. Again, analyzing the reflectors within the volume instead of by bin might solve this confusion. Is this observation based on the north Pacific? Do the authors expect any fossilized slabs in this region based on past subduction?

- The authors claim their technique is very sensitive to reflectors around the size of the cap. If this is the case, I do not understand why overlapping, shifted, bins aren't used. Now, observations are only made in the case that a reflector is sufficiently located within one bin, and are sensitive to where the bin centers are chosen.

- Also, when it comes to discussing resolution, the authors use a value of 3.5 km/s. The shear wave velocity in the mid-mantle is at least 6 km/s. With their estimate, the authors are overestimating their resolution by almost a factor of 2.

- Figure 5c needs a legend, or maybe a logarithmic scaling for the dots (as they become hard to see).

- Page 14, end of first paragraph. 'not present in individual seismograms' -> 'not visible in individual seismograms'.

- As for my earlier point on references 22, and 32. Why is a paper on post-perovskite phase change referenced to discuss a deflection at 660 (the '660' is all but mentioned 4 times in the referenced paper. I think there are much better papers on the deflection at the 660). Reference 27 is about the 660 beneath Iceland, and although that is a reflector in an upwelling region, I don't think it is the reflector the authors mean to refer to here. Maybe the authors meant to cite Jenkins et al. 2017 instead? And yes, these are just two random ones I noticed, so I do suggest the authors check carefully if they mean to cite the papers they do.

Response to Reviewer 1

The manuscript has improved significantly since the original submission and I very much appreciate the time and effort that the authors have invested in addressing the comments from both reviewers. The manuscript should be suitable for publication once the major point is addressed.

Major point:

1. Both reviewers in the original review indicated that the correlation between reflectors and mantle domains was not well-established. To address this, the authors addressed data coverage, shown in Supplementary Table S1. This table shows the data coverage for each domain as well as the ‘proportional coverage’, which measures the degree to which the data coverage mirrors the distribution of fast, neutral, and slow regions in the clustered tomography models.

a. This seems a reasonable start, but stops short of performing a test for statistical significance. It remains unclear whether the correlation between fast/neutral/slow mantle domains and precursors is robust.

We have now performed statistical analysis to determine whether the difference between the observations and their polarity is significant between the different seismic domains, and show that there are statistically significant differences in the combined bin sizes, as well as the 5°, 7.5°, and 10° bins. This is achieved by performing a z-test for observations and non-observations, and for positive and negative polarities, comparing two seismic domains. This method determines whether two seismic domains are significantly different from each other, but not from the third. For example, the observations from fast and slow domains may differ significantly, while the observations from fast and neutral domains do not.

We perform this analysis using a different method to how we previously assigned seismic domains to bins. We believed the previous method, whereby we assigned the seismic domain based on the votes at only the mid-point, was an over-simplification. We have now computed the average proportion of fast/slow/neutral votes per bin, to generate results that reflected the actual composition of each bin. For the bins with reflectors, we obtained the average votes at the observed depth of the reflectors. For bins with no observed reflectors, we obtain the average votes over the full depth range. This then generates a representative of each bin. Rather than assigning a specific domain to the bin, we assign a quantity of votes for each seismic domain, totalling five. We include those results in the Supplementary Information. We have changed Table 1 to correspond to these new results, and updated the relevant parts of the manuscript and Supplementary Information to clarify how we calculate the votes. In the Results section:

“We find statistically significant differences ($p < 0.1$) between seismic domains for bin sizes up to 10°. We characterise each bin according to the average seismic domain votes, and use a z-test to perform systematic statistical comparisons for reflector quantity and polarity between each domain types (Tables 1, S1-S2; see Supplementary Material for full details). For the combined bin approach and 5° bins, the quantities of reflectors differ significantly between seismic domains ($p < 0.05$). As bin size increases, the differences and significance decrease, and are ultimately no longer statistically significant at 15° bin sizes. This statistical analysis further highlights the averaging effects for larger bins.”

In the Methods section:

“Using cluster votes of five different tomographical models (38), we calculate the average vote for the seismic domains across the full lateral extent of each cap at the depth of the reflector, and assign the bin the average votes (totalling five)”

The caption for Table 1, and relevant proportions of the Supplementary Information have

been updated. Numbers in the text from Table 1 have also been updated.

We perform z -tests to determine whether each of the domains are significantly different from the others. The corresponding p -values are presented in Tables S1. To aid the reader, we show the significance in the results Table 1 by using superscript letters (f, s, and n). The presence of a superscript letter means that the two domains is significantly different. This has been added to the table caption.

The description of how we perform the statistics has been added to the Supplementary Information, Correlation to seismic velocity domains section as follows:

“We characterise the reflectors according to their seismic tomography domain. Across each bin, we calculate the proportion of votes for each seismic domain. For bins with reflectors, we evaluate the average of the votes at the depth of the observed reflector, across the bin. For bins with no reflectors, we average the results of the cluster votes over the entire depth range of 800 – 1300 km, at intervals of 50 km. The bin is assigned the average votes for each seismic domain, totalling five per bin (Tables 1 and S1). Note that all of the bins with reflectors necessarily display no reflectors at the majority of depths explored, and so our observations and statistics are heavily skewed towards and characterised by the detections rather than non-observations.

We calculate the statistical significance between two of the seismic domains, comparing the quantity of observations and their polarity. We employ a one-tail z -test, to obtain the probability that the observations and polarities from two types of seismic domain are significantly different from one another (Table S2). This allows us to determine whether one seismic domain is significantly different from the other two, or whether all three differ.

Differences between the domains are significant for the combined bins, and bin sizes up to 10° . The variations become smaller with increasing cap size due to averaging. Thus, we can surmise that statistically significant difference exist between the domains on length scales of the order of 1000 km or less. Averaging effects mean that we do not observed the signatures of each seismic domain in the largest bin sizes, and this is reflected in the statistical probability of a significant difference. This analysis also serves to provide a limit on the useful resolution of this technique for studying the mid-mantle.”

We also reference the decrease in significance with increasing bin size in the discussed of data sensitivity:

“This is corroborated by reflector properties tending towards an average as bin size increases, with differences between domains no longer significant for the largest bins. Analogously, the absence of a global mid-mantle discontinuity is not inconsistent with the widespread presence of regional reflectors.”

b. An alternative explanation for the lack of precursors in fast regions might be that the (presumably compositionally distinct) reflectors do not have as large an impedance contrast relative to their surroundings in these regions, making detection more difficult.

We agree, and this is stated as some alternative reasons for non-observation in the General Observations subsection of the Results section as follows:

“A lack of reflector may result from multiple factors, not limited to the absence of sub-horizontal mid-mantle heterogeneity. For example, small impedance contrasts, gradual transitions (>65 km) including gradual thermal gradients, or complex 3-D structure that does not stack coherently within the bin would not generate reflectors (42-44). Due to the mid-point stacking technique, any reflectors that are not oriented sub-horizontally, such as dipping structures, will not stack coherently.

We have added discussion to various relevant sections, to explicitly state this additional explanation for non-observations in fast regions. We have included the following in the Abstract:

“1) near absence of reflectors in seismically fast regions, likely related to dominantly sub-vertical heterogeneous slab material or small impedance contrasts, both unresolved by our data”.

We also included the following in the Discussion section:

“Alternatively, no reflectors would be detected if the impedance contrasts are small (less than approximately 0.7% averaged across the entire bin).”

Minor points:

2. The authors argue for a compositional origin of the reflectors, which is certainly a reasonable possibility. However it is never established in the paper that a sufficiently large thermal gradient is impossible. Some types of reflectors, particularly the tops of the LLVPs, would have both thermal and chemical contributions.

We have included a calculation in the Data Sensitivity subsection of the Methods section, where we estimate the maximum radial sensitivity of our data:

“The sensitivity of SS data to horizontal discontinuities is estimated using the wavelength λ of our filtered data. Discontinuities which occur over radial length scales of more than $\lambda/4$ cannot be detected. For our data with periods of 15 – 50 s, this corresponds to approximately 65 km. To calculate this length scale, we use the central value of the frequency range (23 s) with a mantle wave velocity of 6 km s^{-1} .”

We also state here:

“Thermal gradients occur on vertical length scales on the order of hundreds of kilometres, and thus are too gradual to observe using SS precursors.”

We have now included a sentence here to incorporate the reviewer’s comments:

“Any extremely large thermal gradients may help to generate reflectors; for example, the tops of LLSVPs may have some thermal contribution, although such a gradient must occur across a vertical distance of less than 65 km.”

We have also added the following to the Discussion section, in which we discuss the slow regions:

“Some thermal contribution may arise if the gradients are extremely strong (occurring over vertical distances of less than approximately 65 km).”

3. As pointed out by reviewer #2, the discussion places significant emphasis on processes observed to occur around 1000 km depth. This is somewhat at odds with the depth histogram of reflectors (Figure 4) which shows a peak around 875 km. This is a similar depth to where Boschi and Becker 2011 find that introducing decorrelation in tomographic models improves model fit to traveltimes.

We have removed much of the emphasis on 1000 km. For example, in the introduction, we have changed one instance to “mid-mantle depths”. We have kept 1000 km in some sentences, such as when referring to stagnation of subducted slabs at 1000 km. The first sentence of the Abstract has also been updated to read:

“Seismic tomography indicates that upwelling and downwelling mantle flow is commonly deflected in the mid-mantle.”

The discussion section had already been updated to focus on the entire mid-mantle depth range.

We have added a reference to Boschi and Becker (2011).

Response to Reviewer 2

I appreciate a number of the clarifications and a more toned down version of this paper. However, I still have a number of issues and questions that either haven't been clarified sufficiently (sometimes due to misunderstandings), or that have arisen from this new version. Specifically now that the tables of observations across the caps and coverage of the domains are added, it allowed me to better assess the statistical significance of the claims made (which unfortunately do not appear significant). I do not think the paper should be accepted in the current state, I encourage the authors to either do more convincing statistical analysis of the claims they make, or re-asses the message they want to give based on their observations.

- The authors claim to do 'statistical analysis' on their results, although this largely consists of giving a lot of percentages to the reader. The table is somewhat overwhelming to the reader (I do appreciate that I asked for analysis of the different bins in the first review). It would be really helpful if the authors put the values in bold, which are statistically significant from the rest? E.g. in the first line 19/21 true observations in the slow domain is statistically significant from the others, while in the 10 degree caps, the observation of 7/11 in the slow domain is not statistically significant from the other domains. Equally, in the first line the 71% of observations in the fast domain is not statistically significant from the neutral domain, and should not be sold as such in the paper. I am afraid, very little in this table stands out as statistically different.

We have now performed statistical analysis to determine whether the difference between the observations and their polarity is significant between the different seismic domains, and show that there are statistically significant differences in the combined bin sizes, as well as the 5°, 7.5°, and 10° bins. This is achieved by performing a z-test for observations and non-observations, and for positive and negative polarities, comparing two seismic domains. This method determines whether two seismic domains are significantly different from each other, but not from the third. For example, the observations from fast and slow domains may differ significantly, while the observations from fast and neutral domains do not.

We perform this analysis using a different method to how we previously assigned seismic domains to bins. We believed the previous method, whereby we assigned the seismic domain based on the votes at only the mid-point, was an over-simplification. We have now computed the average proportion of fast/slow/neutral votes per bin, to generate results that reflected the actual composition of each bin. For the bins with reflectors, we obtained the average votes at the observed depth of the reflectors. For bins with no observed reflectors, we obtain the average votes over the full depth range. This then generates a representative of each bin. Rather than assigning a specific domain to the bin, we assign a quantity of votes for each seismic domain, totalling five. We include those results in the Supplementary Information.

We have changed Table 1 to correspond to these new results, and updated the relevant parts of the manuscript and Supplementary Information to clarify how we calculate the votes. In the Results section:

“We find statistically significant differences ($p < 0.1$) between seismic domains for bin sizes up to 10°. We characterise each bin according to the average seismic domain votes, and use a z-test to perform systematic statistical comparisons for reflector quantity and polarity between each domain types (Tables 1, S1-S2; see Supplementary Material for full details). For the combined bin approach and 5° bins, the quantities of reflectors differ significantly between seismic domains ($p < 0.05$). As bin size increases, the differences and significance decrease, and are

ultimately no longer statistically significant at 15° bin sizes. This statistical analysis further highlights the averaging effects for larger bins.”

In the Methods section:

“Using cluster votes of five different tomographical models (38), we calculate the average vote for the seismic domains across the full lateral extent of each cap at the depth of the reflector, and assign the bin the average votes (totalling five)”

The caption for Table 1, and relevant proportions of the Supplementary Information have been updated. Numbers in the text from Table 1 have also been updated.

We perform *z*-tests to determine whether each of the domains are significantly different from the others. The corresponding *p*-values are presented in Tables S1. To aid the reader, we show the significance in the results Table 1 by using superscript letters (f, s, and n). The presence of a superscript letter means that the two domains is significantly different. This has been added to the table caption.

The description of how we perform the statistics has been added to the Supplementary Information, Correlation to seismic velocity domains section as follows:

“We characterise the reflectors according to their seismic tomography domain. Across each bin, we calculate the proportion of votes for each seismic domain. For bins with reflectors, we evaluate the average of the votes at the depth of the observed reflector, across the bin. For bins with no reflectors, we average the results of the cluster votes over the entire depth range of 800 – 1300 km, at intervals of 50 km. The bin is assigned the average votes for each seismic domain, totalling five per bin (Tables 1 and S1). Note that all of the bins with reflectors necessarily display no reflectors at the majority of depths explored, and so our observations and statistics are heavily skewed towards and characterised by the detections rather than non-observations.

We calculate the statistical significance between two of the seismic domains, comparing the quantity of observations and their polarity. We employ a one-tail *z*-test, to obtain the probability that the observations and polarities from two types of seismic domain are significantly different from one another (Table S2). This allows us to determine whether one seismic domain is significantly different from the other two, or whether all three differ.

Differences between the domains are significant for the combined bins, and bin sizes up to 10°. The variations become smaller with increasing cap size due to averaging. Thus, we can surmise that statistically significant difference exist between the domains on length scales of the order of 1000 km or less. Averaging effects mean that we do not observed the signatures of each seismic domain in the largest bin sizes, and this is reflected in the statistical probability of a significant difference. This analysis also serves to provide a limit on the useful resolution of this technique for studying the mid-mantle.”

We also reference the decrease in significance with increasing bin size in the discussed of data sensitivity:

“This is corroborated by reflector properties tending towards an average as bin size increases, with differences between domains no longer significant for the largest bins. Analogously, the absence of a global mid-mantle discontinuity is not inconsistent with the widespread presence of regional reflectors.”

Additionally, one could account how much is sampled on the global scale. E.g. in the 7.5 degree bin, the 4 (100%) positive observations, might seem significant, but from the supplementary material I gather these 4 observations only sample 8% of the global domain, making it not statically significant claim as the samples size is small.

The 8% (7.7% in Table S2, previously Table S1; values now updated) is the proportional data coverage i.e. the coverage of the slow domain with respect to the other two domains. This corresponds well to the proportion of the slow domain globally, showing that we are not oversampling one domain with respect to the others. The slow domains only represent approximately 8% of the total, not 8% of the slow domain.

The data coverage varies depending on cap size, up to 60-70% for the 15° bins. This is why we prefer the combined bin approach. We have added some clarification of this to the Data Sensitivity subsection as follows:

“Conversely, smaller cap sizes are too noisy in many regions, and suffer from poor data coverage in some areas.”

“Although this approach generates a greater number of bins in regions with the highest data density, we prefer it as it allows us to generate higher resolution imaging where possible, and provide greater global coverage than one bin size alone.”

We have also recalculated the data coverage of each domain, in Table S1 (now Table S2). We use the data coverage for each bin size, and assign a position and depth the seismic domain for which there are three votes or more. This provides much more accurate data regarding sampling. We now show the data coverage of the domain with respect to the global coverage of that domain, rather than total global coverage, which is clearer for the reader (and the phrasing in Table S2 caption is clearer also). For the 7.5° bins, for example, Table S2 shows that we sample 31.5% of the slow domains globally, and slow domains make up 7.8% of our coverage for these bin sizes. Thus, actually we sample 1/3 of the slow regions available to sample. For the combined bins, which we use for most of the analysis, the data coverage is in the high 60% range across all bins.

Related to this point, the discussion starts off with the following sentence: *'The observed mid-mantle reflectors exhibit no geographic relationship to surface features; instead they correlate to mid-mantle structure as imaged by seismic tomography.'* In the previous chapter, the author has just shown that they do not correlate to seismically imaged vertical gradients, so I presume the authors mean there is correlation with the defined seismic domains here. In the table below, I show that when adding up all the domains, there is no significant difference of scattering within the domains. The number of reflectors observed in each domain, as well as the number of positive reflectors in each domain, correlate with the amount of bins assigned to that domain, showing now significant differences between the domains.

A problem with the way that the reviewer has made this table is that some regions are now counted four times over, such as the central Pacific. The results produce an average across the lateral variations, and hence are misleading. This is why we use the combined bins – populating with the smallest cap sizes, then ensuring as we iteratively populate the map, we do not add bins which overlap more than to the centre of the bins already populated (i.e., how the original bins

All caps combined	Fast	Neutral	Slow
range proportional data coverage across the different caps	19-24%	68-70%	7-12%
Observations	63	20	30
% of total observations	21.4%	68.4%	10.2%
Positive observations	47	144	18
% or positive observations	22.4%	68.9%	8.6%
non-detections	29	99	13
% of non-detections	20.5%	70.2%	9.2%

were located for each cap size).

A second problem with adding up the values from the caps like that is that we show in the paper that there is variation between cap sizes, due to the lateral variation. The differences between the regions vary across the different cap sizes.

Also, there are a few errors, probably typos; for example, total neutral observations are 202 not 20, and neutral non-detections are 109, not 99. This may have skewed the analysis. We have now performed statistical analysis, as suggested by both reviewers; the results for this are outlined in the response to the first point.

If the authors want to build their claims on statistics, I think they should do it properly. If they want to make sweeping, qualitative statements (like is quite common in the field), don't claim to do a robust analysis and move away from words like 'correlate' and 'statistical analysis' (although I wouldn't suggest going this route). In light of broad statements, I think the observations in this paper show, dense observations of horizontal reflectors beneath the central Pacific (in the 5 degree bins), and observations of broad reflectors around 850 and 1000 km.

We agree with the reviewer's comments and have therefore performed statistical analysis. Our results are described in the response to the first point.

We have included reference to the dense horizontal reflectors in the central Pacific, and broad reflectors around 850 and 1000 km, as follows:

“The geographically most extensive reflectors are located beneath the Pacific Ocean and (offshore) eastern South America.”

“The South Pacific is our best-resolved example of a slow region, with dense horizontal reflectors that vary on short lateral length scales.”

“In our best-example “neutral region” in the Northeast Pacific, there are two dominant reflectors at 850 and 1,050 km, with scattered deeper detections (Figure 6).”

- The authors show a synthetic test of potential rotation of the components at the station. In this test they show energy leakage at a time and slowness, which they would interpret in their stacks as coming from a mid-mantle reflector. It is unclear to me what phase from the radial is leaking in here and the authors do not mention this (the timing and slowness of the observed arrival on the black line of predicted reflectors does not seem to correspond with the predictions in the table provided in the rebuttal). I am not so much worried about consistent rotation of the horizontal components at all the stations used, but a bias of the upper mantle anisotropy (which can be consistent over large regions and many stations) could cause leakage between the components. The authors claim to throw out similar slowness stacks due to distortion on the S660S phase,

however I don't find the rotated S660S arrival in the synthetic more distorted than in the stacks shown in Figures 2a and 2b. I understand that stacking the radial component for all the bins and analysing them is beyond the scope of this research. However, I don't think it is too much to ask to stack the radial component for one good quality stack with an identified arrival and show that there is no interfering energy on the radial component.

We picked the radial component data for the bin at 30°N , 208.6°E , which is shown in Figure 2a. This is a 10° bin, with one clear S1000S arrival in the precursor window, and selected because it is high quality, shown in the manuscript, and has a large number of data. The transverse component data has 1492 records used to make the vespagram. For the radial component, we began with an initial dataset of 41,995 seismograms. This was reduced to 5,114 seismograms following quality checks, and of these, 1595 records were picked and stacked. The data were processed as the transverse: in displacement, filtered for 15 – 75 s, aligned on the SS theoretical time, quality checked, and then hand-picked on SS. After picking, the original data were then aligned to the pick and then filtered for 15 – 50 s, and then normalised and aligned automatically to the maximum peak of the SS pick. The stack for a filter of 15 – 50 s (the same as in Figure 2 in the paper) are shown below (Figure 1). There are no arrivals with the correct arrival time and slowness of SS precursors, in the mid-mantle window. The energy arriving at these times on the cross-section corresponds to that leaking from higher or lower slownesses. Note that the arrival in Figure 2 in the manuscript is a peak-trough-peak shape, with the trough arriving at around 305 s earlier than SS. Thus, we therefore conclude that there is no interfering energy leaking from the radial component, in the transverse vespagrams that we show.

We do not add this figure to the manuscript, since this analysis is extremely preliminary and incomplete. One reason for this is that our picking process is by event, so that we can identify the waveform of SS. The SS waveform can vary significantly for data from different events, but is generally consistent within an event. This allows us to ensure our picks are consistent. Picking by location removes this aid, and probably means that we discarded more data than needed. A corresponding global radial component dataset will be the focus of a future study, of which (an improved version of) the data in this vespagram will be part.

Figure 1. Radial component vespagram for 30°N, 208.6°E, showing no energy arriving at mid-mantle precursor times or slownesses. Energy which is seen on the cross-section is leaking from higher or lower slownesses, and thus does not correspond to SS precursors.

- From the rebuttal to point 4.a, I gather that the authors switched readily between analyses the observations of the lateral bins (e.g. to determine the domain, and fill in the table), as well as a function of volume (e.g. when drawing the conclusion that there is a lack of scatterers in the fast domain). This is very confusing for the reader now, as the latter conclusion is not clear from the table or figures. Why do the authors not do all their analysis in volume space, and also let the domains vary as a function of depth? (Maybe some of the analyses would become statistically interesting in volumes space).

Fast domains were just used as an example in the response to the review, because that was mentioned in the original review comment. We do not switch back and forth between lateral bins and volume space – volume space alone is used for the non-observations. The alternatives are to generate a lateral bin every 50 km, which would skew the results towards non-observations; or, to select an arbitrary mid-mantle depth to use for assigning domains to non-observations, which may be unrepresentative of the bin.

Although we could perform similar analyses using volume space for the observations, that would remove some information from our results, specifically the depth at which the reflector is detected. Alternatively, if the reviewer means to consider velocity variation across depths, we already did that in the form of considering velocity gradients from shear wave tomography models (Supplementary Figure 16). Note that we now do consider the entire lateral bin with how we define and assign the seismic domains.

- The point 4.b.i I made in the first review, might have been misunderstood, as the authors explain in the rebuttal that the Central Pacific is well covered by the data. I totally agree it is well-sampled. With my point, I meant that it was over-sampled in the combined caps approach, where this area is reflected by dense grid of degree 5 caps. From table 1, I now gather that this claim of lots of observations in the slow domain is entirely biased by the 5 degree caps, which are densely sampling the mid-Pacific in the Combined caps approach. In non of the other degree caps does the slow area stand out as significantly different. In the light of the discussion that each of the various degree caps has sensitivity to different size of reflectors, the combined approach seems to not add much value.

Point 4.b. stated:

'reflections from the top of seismically-slow thermochemical piles beneath the Pacific;'

i. With regards to this claim, I am most worried about the uneven sampling of the bins. Only very little of the slow region (mainly central Pacific) is sampled, and this part is very densely sampled.

We interpreted this comment as stating that only very little of the slow region is well-sampled, in the central Pacific. We then show that most of the seismically-slow thermochemical pile region in the rest of the Pacific is sampled.

The combined approach shows lateral variation on the length scales of the smaller bins. Thus, the reflectors in the Pacific region can be seen to vary on the length scales of the smallest bins, but not the largest, providing information on the length scale of heterogeneity. We state this in the General Observations subsection of the Results section as follows:

“We examine these averaging effects for larger bins to confirm variation in reflector coherency across lateral length scales (Figures S5, S6). The averaging effect is exemplified beneath the mid-Pacific Ocean, where the depths of reflectors vary significantly for the 5° bins (Figure S5a), while the more averaged 15° bins predominantly display fewer reflectors (Figure S5d, S7).”

We added the following to the Methods section, Data Sensitivity subsection:

“Although this approach generates a greater number of bins in regions with the highest data density, we prefer it as it allows us to generate higher resolution imaging where possible, and provide greater global coverage than one bin size alone.”

We also added the following to the caption of Figure 3:

“The data coverage (non-hatched areas) provides an indicator as to the sensitivity of each bin.”

Furthermore, we have included corresponding figures showing each bin size individually to the Supplementary Information.

- Figure 6 is a lot for the reader to take in, and the colorbar is given limited explanation. Now that the authors have clarified that they define their domains at 3 votes, would it not be easier to just plot the domains in the background (and one color across the whole depth range, if the authors assign a domain to each bin), instead of the full triangle of colors. This can also clarify what region is not counted as one of the domains (do any of the caps fall within a boundary region where votes don't agree? From the table in the supplementary material I gather all the domains add up to 100%. Or are observations in regions with fewer than 3 votes for one domain discarded?).

The point of Figure 6 was to show a cross-section through the clustering models, thus assigning the entire depth range as one domain would not make sense in terms of the context of the figure. This is particularly true given how the Figure is referenced in the manuscript:

“Reflectors are absent in only very few slow domain bins (primarily beneath the Pacific Ocean),

and occur near the edges or tops of slow domains (Figure 6), or LLSVPs.”

“In our best-example “neutral region” in the Northeast Pacific, there are two dominant reflectors at 850 and 1,050 km, with scattered deeper detections (Figure 6).”

Removing the tomographical background would essentially make the figures into a poorer projection of Figure 3a.

The colour scale is selected to correspond to that used in Cottaar and Lekic (2016). We agree that this may be confusing for readers unfamiliar with the paper. To make the interpretation of the colour scale easier for the reader, we now use the same background colours and meanings as the observation map in Figure 3. It is now straightforward to see the domains in which the observations fall.

We also added more explanation to the figure caption, to aid the reader:

“Cross-sections through vote maps for five mantle tomography models (38) across four regions of high data density. Our observations are superimposed at their calculated depths. As in Figure 3, the background colours correspond to the average tomographically fast (blue) and slow (pink) regions, calculated at every 50 km depth. Unshaded regions are neither fast nor slow.”

Observations which fall in regions with fewer than 3 votes are negligible (two in the 10° and 5° caps, one in the 15° and 7.5° caps), and were previously discarded for the purposes of the statistics. Now, we do not discard any bins, since we take the average properties.

- The first conclusion in the abstract states '1) near absence of reflectors in seismically fast regions, likely related to dominantly sub-vertical heterogeneous slab material; '. I've already mentioned that I don't first half of the claim is not statistically significant, and I don't see how the second half follows on the first. Maybe state '1) near absence of broad reflectors in seismically fast regions, although previously observed small-scale sub-vertical heterogeneous slab material would not be imaged by our dataset; '. Related to this, it might be helpful to give the length-scale of potential observations in the abstract, so the reader know what type of reflectors are being referred to.

We have now shown that the first half of the sentence is statistically significant. The second half follows from the first in terms of that we expect heterogeneity in fast regions, but the data cannot resolve it, and so thus the heterogeneity must be of an orientation to which SS precursors are not sensitive. We have changed the sentence to add clarification as follows:

“near absence of reflectors in seismically fast regions, likely related to dominantly sub-vertical heterogeneous slab material or small impedance contrasts, both unresolved by our data”.

(the reference to small impedance contrasts relates to a comment from Reviewer 1).

We have also included lengthscales of observations as follows:

“Here, we perform a systematic global scale interrogation of mid-mantle seismic reflectors, with lateral length scales of 500-2,000 km, in the depth range of 800–1,300 km.”

- It is also very confusing for the reader that in the cartoon that one blue bit is fast domain, and the other is neutral domain (and it appears both reviewers were confused by this). Maybe fossilized, neutralized, slab material should not be drawn as blue or connected to the surface. Again, analyzing the reflectors within the volume instead of by bin might solve this confusion. Is this observation based on the north Pacific? Do the authors expect any fossilized slabs in this region based on past subduction?

We agree and we have changed the cartoon as suggested. We have also changed the analysis to take into account the average properties across the bin, rather than just defining the domain by

the mid-point. The cartoon is a simplified conceptual interpretation of our observations, generalising by domain.

- The authors claim their technique is very sensitive to reflectors around the size of the cap. If this is the case, I do not understand why overlapping, shifted, bins aren't used. Now, observations are only made in the case that a reflector is sufficiently located within one bin, and are sensitive to where the bin centers are chosen.

We do use overlapping bins. From the Section sub-section of the Methods section:

“We partition the data into overlapping spherical caps based on their bounce points, to generate regional maps.”

We have added the following to the next sentence for clarification:

“The geometry is such that the centre-point of a bin corresponds to the edge of each adjacent bin.”

- Also, when it comes to discussing resolution, the authors use a value of 3.5 km/s. The shear wave velocity in the mid-mantle is at least 6 km/s. With their estimate, the authors are overestimating their resolution by almost a factor of 2.

This was a typo. We have changed the value, and updated the calculation and all included numbers to provide a more exact value of 65 km.

- Figure 5c needs a legend, or maybe a logarithmic scaling for the dots (as they become hard to see).

We changed the dots to show ranges of bin sizes (in increments of 200 data points), rather than corresponding directly to the amount of data per bin. This means that the smallest bins can now be seen, but does not significantly alter the figure. We added a key showing the quantity of data which corresponds to the circle sizes. We also changed the corresponding figure in the supplementary material, and added a key to this figure too.

- Page 14, end of first paragraph. 'not present in individual seismograms' -> 'not visible in individual seismograms'.

This has been changed.

- As for my earlier point on references 22, and 32. Why is a paper on post-perovskite phase change referenced to discuss a deflection at 660 (the '660' is all but mentioned 4 times in the referenced paper. I think there are much better papers on the deflection at the 660). Reference 27 is about the 660 beneath Iceland, and although that is a reflector in an upwelling region, I don't think it is the reflector the authors mean to refer to here. Maybe the authors meant to cite Jenkins et al. 2017 instead? And yes, these are just two random ones I noticed, so I do suggest the authors check carefully if they mean to cite the papers they do.

We have changed reference 22 (now 23) to Tosi & Yuen (2011). Jenkins et al. (2017) is reference 33. We have checked the references to all other papers throughout, and ensured that the correct numbers are linked and correct papers referenced. Note some references have now been altered.

Reviewers' comments:

Reviewer #2 (Remarks to the Author):

I think the comments of the reviewers have been sufficiently addressed, and the statistical analysis in the paper has significantly ($p < 0.05$) improved.

There are two bits of wording that I suggest the authors change:

- 'with fewer observational peaks around 1000, 1100 and 1200 km' (first paragraph, page 5). Maybe say something like 'with less pronounced peaks in the range of 1000-1300 km.'
- 'Fast domains are characterized by a dominance of non-detections' (last paragraph, page 8). Change 'dominance' to 'relatively more'. Maybe discuss fast domains after the slow domain to compare the relative values.
- The percentage values in the discussion are maybe outdated. I assume they meant to represent the combined cap values in table 1.

Reviewer #3 (Remarks to the Author):

I had a fresh look at manuscript NCOMMS-17-01274B by Waszeck et al, 2017. This paper has already gone through a few rounds of reviews and it seems that the authors have already done an extensive work in answering a large number of detailed comments by two reviewers. I will therefore not enter in a detailed review of all the technical aspects of the paper, but I will give some general comments.

First, this work is a nice study and probably deserves publication in a specialized journal.

Second, a large part of the paper is very technical (I understand that it probably results in part from the two rounds of reviews). I agree with reviewer 2 that the reader is somewhat overwhelmed by the large number of figures (6 in the main paper, 16 in the supplementary material) and tables (3 tables, each with a large number of information). In the "Results" section, the authors often refer to two and sometimes up to four figures or Tables at the same time, sometimes asking in addition to look at the Supplementary Material (for example, page 8 the reader is asked to look at the same time at Tables 1, S1 S2 and supplementary Material, and if you try the exercise, you will probably acknowledge that it takes some time to understand what's going on). This is very disturbing for the reader. One part of the problem might be that 1) a lot of figures are in supplementary material; 2) the "methods" section is not included in the main text. This is ok, when it is possible to understand the results without going too much into the technical details. However, it seems to me that for this paper, it is difficult to focus on the results and bypass technical aspects (for example, the statistical significance of the results is a key question here). For this reason, it seems to me that a letter format may not be well suited for this kind of paper. Unless Nature Communication allows publishing results in a different format, I have the feeling that this paper is better suited for JGR or GJI than a letter type journal. I understand that the authors have already put a lot of efforts in this paper. However, they may get a faster review if they send their review+answer to another journal. Note also that I was somewhat disturbed by some percentage numbers (e.g. 64%, 93% 43% and 76% page 9 that I couldn't retrieve in tables), but maybe I missed something?

Finally I agree with other reviewers that the authors may want to think again about the statistical significance of their results. As there is no obvious trend in the data, I'm not sure that pushing the statistical analysis will give more convincing results. The variety in the properties of these reflectors and the fact that some of them may not be detected in global tomography is to me already an important result.

Response to Reviewer 2

I think the comments of the reviewers have been sufficiently addressed, and the statistical analysis in the paper has significantly ($p < 0.05$) improved.

Thank you, we appreciate your input.

There are two bits of wording that I suggest the authors change:

- 'with fewer observational peaks around 1000, 1100 and 1200 km' (first paragraph, page 5). Maybe say something like 'with less pronounced peaks in the range of 1000-1300 km.'

This has been changed.

- 'Fast domains are characterized by a dominance of non-detections' (last paragraph, page 8). Change 'dominance' to 'relatively more'. Maybe discuss fast domains after the slow domain to compare the relative values.

We changed the phrasing as suggested, and rearranged the section. We now discuss the observations from slow domains, then fast, then neutral; this now corresponds to the order in the discussion. We have made some editing to the results paragraphs in order to provide more comparison between fast and slow domains.

- The percentage values in the discussion are maybe outdated. I assume they meant to represent the combined cap values in table 1.

Thank you for pointing this out. We have checked and updated the outdated values in the main text accordingly.

Response to Reviewer 3

I had a fresh look at manuscript NCOMMS-17-01274B by Waszeck et al, 2017. This paper has already gone through a few rounds of reviews and it seems that the authors have already done an extensive work in answering a large number of detailed comments by two reviewers. I will therefore not enter in a detailed review of all the technical aspects of the paper, but I will give some general comments.

First, this work is a nice study and probably deserves publication in a specialized journal.
Thank you, we appreciate your input.

Second, a large part of the paper is very technical (I understand that it probably results in part from the two rounds of reviews). I agree with reviewer 2 that the reader is somewhat overwhelmed by the large number of figures (6 in the main paper, 16 in the supplementary material) and tables (3 tables, each with a large number of information). In the "Results" section, the authors often refer to two and sometimes up to four figures or Tables at the same time, sometimes asking in addition to look at the Supplementary Material (for example, page 8 the reader is asked to look at the same time at Tables 1, S1 S2 and supplementary Material, and if you try the exercise, you will probably acknowledge that it takes some time to understand what's going on). This is very disturbing for the reader. One part of the problem might be that 1) a lot of figures are in supplementary material; 2) the "methods" section is not included in the main text. This is ok, when it is possible to understand the results without going too much into the technical details. However, it seems to me that for this paper, it is difficult to focus on the results and bypass technical aspects (for example, the statistical significance of the results is a key question here). For this reason, it seems to me that a letter format may not be well suited for this kind of paper. Unless Nature Communication allows publishing results in a different format, I have the feeling that this paper is better suited for JGR or GJI than a letter type journal. I understand that the authors have already put a lot of efforts in this paper. However, they may get a faster review if they send their review+answer to another journal.

Nature Communications clarified that we are permitted up to 5000 words in the main text, and unlimited words in the Methods section, in addition to up to 10 figures/tables. As per the Editor's suggestion, we have moved all of the text in the Supplementary Material to the main manuscript. We also moved two important figures from the supplement to the main manuscript (the Axisem modelling for observability as a function of size, strength, and bin size; the vertical velocity gradient calculations from 3-D models). Thus, we now no longer need to refer the reader to the Supplementary Material text. Furthermore, this means that the figures in the Supplementary Material are all supplementary to the main figures now; i.e., extra examples of vesperagrams of different quality, or alternative plots of figures in the main text such as observations separated by bin size rather than combined.

We agree that some of the references were lengthy and demanding for the reader. To address this in the first instance, we have worked through the manuscript to find locations in which we refer to multiple figures or tables, and have rephrased these sections (the sentence on page 8 which the reviewer mentioned has been rewritten now, for instance). Some of the locations with multiple references were previously written in that way to include every possible relevant figure. As an example, in the Methods Data and processing subsection, we referred to every single vesperagram when discussing stacking. Instead, we now clarify that these are various examples of

vespagrams. In other locations, particularly where we refer to Supplemental Figures, we add in extra description to explain their relevance. Furthermore, we have made use of the word limit in order to add extra information in the main text, so that it can stand alone and the reader does not need to refer as much to the Methods section in order to understand the main results. In terms of the technical details, adding this extra information to the main text has helped to place it in the context of the wider results, without going into too much technical detail in the main manuscript. We keep some references to the Methods section (with more description) for those readers that are interested in the technical details.

We believe that these edits we have made in response to the Editor's and Reviewer's suggestions have significantly helped to improve the manuscript to aid the reader, and make it much better suited for the Nature Communications format.

Note also that I was somewhat disturbed by some percentage numbers (e.g. 64%, 93% 43% and 76% page 9 that I couldn't retrieve in tables), but maybe I missed something?

Thank you for pointing these out. These were typos that were not fixed in the second round of revisions, after modifying our methods of calculating proportion of observations and positive polarity observations based on the second set of comments from the previous reviewers. We have now updated those.

Finally I agree with other reviewers that the authors may want to think again about the statistical significance of their results. As there is no obvious trend in the data, I'm not sure that pushing the statistical analysis will give more convincing results. The variety in the properties of these reflectors and the fact that some of them may not be detected in global tomography is to me already an important result.

We have retained the statistical analysis as Reviewer 1 and Reviewer 2 believed it was important. We agree that the wide variety of the reflectors, and their absence in global models, is a major finding of the paper. Thus, we have re-worked the manuscript to emphasise this point, and include the statistics to highlight the variation across length scales (and thus, the variation in lateral coherency of the reflectors).